# The modulation of savouring by prediction error and its effects on choice.

Kiyohito Iigaya[1]*, Giles W Story[2,3], Zeb Kurth-Nelson[2,3], Raymond J Dolan[2,3], Peter Dayan[1]

[1]Gatsby Computational Neuroscience Unit, University College London, London, United Kingdom; [2]The Wellcome Trust Centre for Neuroimaging, University College London, London, United Kingdom; [3]Max Planck UCL Centre for Computational Psychiatry and Ageing Research, London, United Kingdom

**Abstract** When people anticipate uncertain future outcomes, they often prefer to know their fate in advance. Inspired by an idea in behavioral economics that the anticipation of rewards is itself attractive, we hypothesized that this preference of advance information arises because reward prediction errors carried by such information can boost the level of anticipation. We designed new empirical behavioral studies to test this proposal, and confirmed that subjects preferred advance reward information more strongly when they had to wait for rewards for a longer time. We formulated our proposal in a reinforcement-learning model, and we showed that our model could account for a wide range of existing neuronal and behavioral data, without appealing to ambiguous notions such as an explicit value for information. We suggest that such boosted anticipation significantly drives risk-seeking behaviors, most pertinently in gambling.

## Introduction

When people anticipate possible future outcomes, they often prefer their fate to be revealed in advance by a predictive cue, even when this cue does not influence outcome contingency. This is usually called 'observing', or 'information-seeking', behavior.

Recently, Bromberg-Martin and Hikosaka (*Bromberg-Martin and Hikosaka, 2009*; *2011*) reported an influential series of studies into the neural basis of the observing behavior of macaque monkeys. They tested subjects' preferences between three targets that were followed by cues that resolved uncertainty about the volume of an upcoming reward (small or large) to different degrees. Subjects strongly preferred a '100% info' target, which was followed by uncertainty-resolving, definitive, cues, over a '50% info' target, which was followed either by definitive cues or by a totally ambiguous cue; and preferred the latter target over a '0% info target', which was always followed by an entirely ambiguous cue. Neurons in lateral habenula responded differently to the same definitive, reward-predicting, cue depending on the target that had previously been chosen (100% or 50% info). The authors concluded that these neurons index what they called 'information prediction errors' along with conventional reward prediction errors (*Schultz et al., 1997*; *Matsumoto and Hikosaka, 2007*), and that biological agents ascribe intrinsic value to information.

A yet more striking finding is that animals appear willing to sacrifice reward for taking advance information. This is known for birds (*Zentall, 2016*; *McDevitt et al., 2016*), monkeys (*Blanchard et al., 2015a*), and humans (*Eliaz and Schotter, 2010*; *Molet et al., 2012*). Extensive studies on birds (mostly pigeons) showed that animals prefer a less rewarding target that is immediately followed by uncertainty-resolving cues, over a more rewarding target without such cues (e.g. 20% chance over 50% chance of reward (*Stagner and Zentall, 2010*; *Vasconcelos et al., 2015*), 50% chance over 75% chance of reward (*Gipson et al., 2009*), 50% chance over certain reward

**eLife digest** People, pigeons and monkeys often want to know in advance whether they will receive a reward in the future. This behaviour is irrational when individuals pay for costly information that makes no difference to an eventual outcome. One explanation is that individuals seek information because anticipating reward has hedonic value (it produces a feeling of pleasure). Consistent with this, pigeons are more likely to seek information when they have to wait longer for the potential reward. However, existing models cannot account for why this anticipation of rewards leads to irrational information-seeking.

In many situations, animals are uncertain about what is going to happen. Providing new information can produce a "prediction error" that indexes a discrepancy between what is expected and what actually happens. Iigaya et al. have now developed a mathematical model of information-seeking in which anticipation is boosted by this prediction error.

The model accounts for a wide range of previously unexplained data from monkeys and pigeons. It also successfully explains the behaviour of a group of human volunteers from whom Iigaya et al. elicited informational and actual decisions concerning uncertain and delayed rewards. The longer that the participants had to wait for possible rewards, the more avidly they wanted to find out about them. Further research is now needed to investigate the neural underpinnings of anticipation and its boosting by prediction errors.

(*Spetch et al., 1990*; *McDevitt et al., 1997*; *Pisklak et al., 2015*). Crucially however, the pigeons only show this preference when the delay, $T_{\text{Delay}}$, between the choice and the reward is sufficiently long. Another salient experimental observation in *Spetch et al. (1990)* is that after choosing a less rewarding, 50% chance, target, some of the pigeons were also seen to peck enthusiastically during the delay following the cue informing them that reward would arrive. By contrast, they were comparatively quiescent during the delay after choosing the certain reward target followed by a similar cue (*Spetch et al., 1990*).

It remains a challenge to account for these data on the preference of advance information. The delay-dependent preference reward predictive cues shown by *Spetch et al. (1990)* cannot depend on conventional Shannon information, since this is normally independent of delay. Furthermore, targets associated with less Shannon information can be more attractive (*Roper and Zentall, 1999*; *Zentall, 2016*). Equally, evidence from the activity of lateral habenula neurons (*Bromberg-Martin and Hikosaka, 2011*) provides no support for the recent suggestion that disengagement caused by uninformative cues could cause the seeking of informative cues (*Beierholm and Dayan, 2010*).

Here, we offer a new explanation of observing and information-seeking behavior that accounts for the effects of delays reported in the pigeon experiments and the effects of changing the probability of reward. We follow an established notion called the *utility of anticipation* (*Loewenstein, 1987*; *Berns et al., 2006*; *Story et al., 2013*). These investigators have shown that subjects consider the delay to a future reward as itself being appetitive (think, for instance, of yourself waiting for an upcoming vacation trip), associated with a positive utility of anticipation. This is often referred to as savouring (the anticipation of negative outcomes is called dread), and coexists with a more conventional effect of delay, namely temporal discounting (*Loewenstein, 1987*; *Loewenstein and Prelec, 1993*; *Schweighofer et al., 2006*; *Kable et al., 2010*).

In this framework, we hypothesize that the level of anticipation can be boosted by the (temporal difference) prediction errors caused by predictive cues that resolve reward uncertainty. Pigeons' vigorous pecking following those cues is a sign of the boost. That is, the definitive reward cue following a partial target evokes a positive (temporal difference) prediction error – the difference between the expected (partial chance) and actual outcome (reward for sure) is positive (*Schultz et al., 1997*). We suggest that the impact of this is to increase savouring. By contrast, the certain target elicits no such prediction error and so, within this account, will not increase savouring. Put simply, our model posits that unexpected news of upcoming pleasant (or unpleasant) outcomes boosts the savouring (or dread) associated with such outcomes.

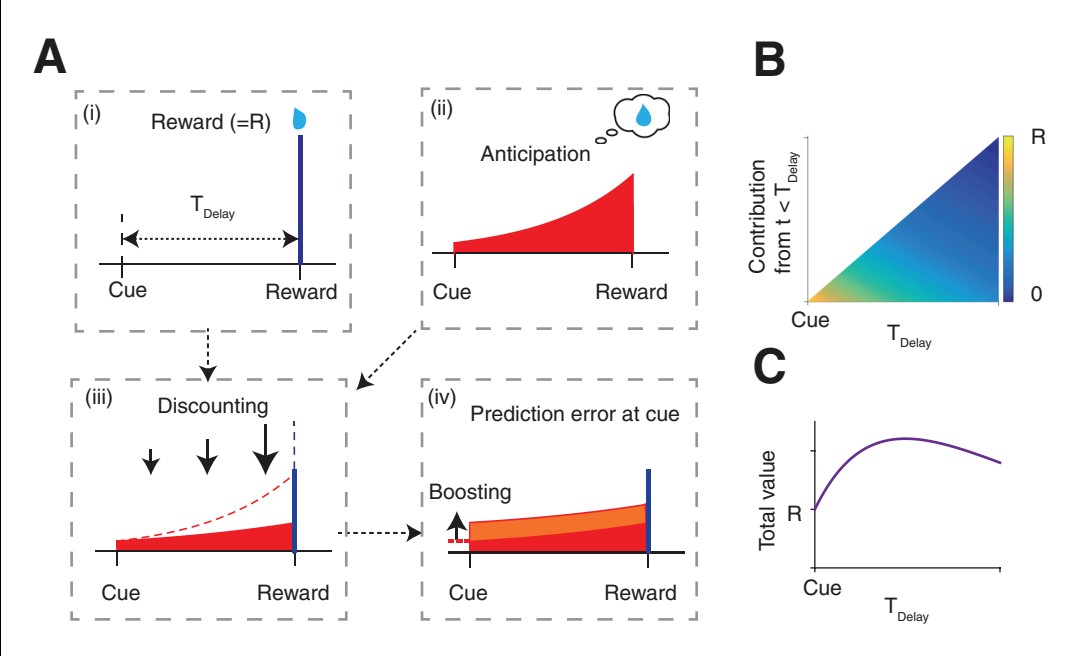

**Figure 1.** The model. (**A**) The value of the cue is determined by (ii) the anticipation of upcoming reward in addition to (i) the reward itself. The two are (iii) linearly combined and discounted; with the weight of anticipation being (iv) boosted by the RPE associated with the predicting cue. (**B**) The contribution of different time points to the value of predicting cue. The horizontal axis shows the time of reward delivery. The vertical axis shows the contribution of different time points to the value of the predicting cue. (**C**) The total value of the predicting cue, which integrates the contribution along the vertical axis of panel (**B**), shows an inverted U-shape.

The hypothesis of boosting yields a parsimonious explanation for observing and information-seeking, consistent with all existing neural and behavioral data, including a range of seemingly paradoxical findings (*Gipson et al., 2009*; *Spetch et al., 1990*; *Stagner and Zentall, 2010*; *Bromberg-Martin and Hikosaka, 2009*; *2011*; *Zentall, 2016*). Here we also conducted human behavioral studies to test the delay $T_{\text{Delay}}$ dependence of observing and information seeking, which has so far only been subject to limited tests in animal studies (*Spetch et al., 1990*).

## Results

### The RPE-anticipation model

Our model relies on the established economic theory called the utility of anticipation (*Loewenstein, 1987*), which proposes that the anticipation of future reward has a positive subjective value which is added to the actual value of the reward. Our formulation follows prior characterizations (*Loewenstein, 1987*; *Berns et al., 2006*; *Story et al., 2013*). Consider the case in which a subject takes an action and receives a pre-reward cue at $t = 0$, and then a reward $R$ at $t = T_{\text{Delay}}$ (*Figure 1A*). The anticipation of the reward is worth $a(t) = Re^{-\nu(T_{\text{Delay}} - t)}$ at time $t$ (*Figure 1A-(ii)*), where $\nu$ governs the rate of growth of this factor. A small $\nu$ means that the anticipation grows gradually in time, while a large $\nu$ means that the anticipation increases steeply near the delivery of rewards.

The value of the pre-reward cue is determined by what follows the cue, which, under conventional temporal difference (TD) learning, would have been the reward itself (*Figure 1A-(i)*), discounted in time with a rate $\gamma$. Here, however, in addition to the reward itself, we have the anticipation of the reward that takes place continuously over time. Thus the total value of the predictive cue, $Q$ is the sum of the discounted reward (blue bar in *Figure 1A-(iii)*), and the temporally discounted anticipation, where the latter is integrated over time from the presentation of the cue up to reward delivery (red area in *Figure 1A-(iii)*).

Note that the integration of anticipation suggests that the total amount of anticipation contributing to the value of the cue can increase as $T_{\text{Delay}}$ is increased. This can be seen in *Figure 1B*, in which the color code indicates the contribution of the temporally discounted anticipation at time $0 < t < T_{\text{Delay}}$ to a predictive cue. The horizontal axis indicates different delay conditions $T_{\text{Delay}}$, and the vertical axis shows the different time points $t$ between the cue ($t = 0$) and the time of reward delivery ($t = T_{\text{Delay}}$). *Figure 1C* shows the total values for different delay length $T_{\text{Delay}}$, which are the integrals of contributions over the vertical axis in *Figure 1B*. As seen, the total value usually takes the maximum value at a finite $T_{\text{Delay}}$, which is larger than the value $R$ of the reward itself (*Figure 1C*). While the actual peak is determined by the competition between the anticipation $\nu$ and discounting $\gamma$, this inverted U-shape was confirmed previously for the case of savoring, using hypothetical questionnaire studies (*Loewenstein, 1987*). (This inverted U-shape holds in general in the model, unless the relative weight of anticipation $\eta$ is zero, or growth and discounting are too steep relative to each other $\nu \ll \gamma$ or $\nu \gg \gamma$. )

Previously, the total value of cue $Q$ has been expressed as a sum of the value of anticipation and the reward itself:

$$Q = \eta\, V^{[\text{Anticipation}]} + V^{[\text{Reward}]} \tag{1}$$

with the relative weight of anticipation $\eta$ being treated as a constant. Here we hypothesized that reward prediction errors (RPE) $\delta_{pe}$ in response to the predictive cue (*Schultz et al., 1997*) can boost anticipation (*Figure 1A-(iv)*). Our proposal was inspired by findings of a dramatically enhanced excitement that follows predictive cues that resolve reward uncertainty appetitively (*Spetch et al., 1990*), which will generate positive RPEs. A simple form of boosting arises from the relationship

$$\eta = \eta_0 + c|\delta_{pe}| \tag{2}$$

where $\eta_0$ specifies base anticipation, and $c$ determines the gain. That anticipation is boosted by the *absolute value* of RPE turns out to be important in applying our model to comparatively unpleasant outcomes, as confirmed in our own experiment. Note that anticipation can only be boosted by the RPE that precedes it – in this case arising from the predictive cue. Any RPE associated with the delivery of reward would have no anticipatory signal within the trial that it could boost. We ignore any subsidiary anticipation that could cross trial boundaries.

## Physiological and behavioral aspects of observing and information-seeking

*Figure 2A,B,C* illustrates the design and results of the experiment mentioned above in which macaque monkeys exhibit observing, or information-seeking, behavior (*Bromberg-Martin and Hikosaka, 2011*). Briefly, there were three targets: 1) a 100% info target that was always followed by a cue whose shapes indicated the upcoming reward size (big or small); 2) a 0% info target that was always followed by a random cue whose shapes conveyed no information about reward size; and 3) a 50% info target that was followed half the time by informative cues and half the time by random cues (*Figure 2A*). *Figure 2B* shows the strong preference the subjects exhibited for the 100% over the 50%, and the 50% over the 0% info targets. *Figure 2C* shows the difference in activity of lateral habenula neurons at the time of the predictive cues depending on the preceding choice of target.

*Figure 2D* shows that our model captured subjects' preferences for each of the targets; and *Figure 2E,F* shows that it also accounted for the different sizes of RPEs to the same reward predictive cues when they followed different info targets (noting that the responses of lateral habenula neurons are negatively correlated with the size of reward prediction errors; *Matsumoto and Hikosaka, (2007)*; *Bromberg-Martin and Hikosaka, (2011)*). The difference in RPE sizes arose from the different values of the targets, which itself arose from the different magnitudes of anticipation associated with the cues. We found that these results held across a wide range of parameter settings (*Figure 2—figure supplement 1*). Note that RPE-boosting is necessary to capture the data, as the simple baseline anticipation model predicted the same level of anticipation following each target, leading to no preference between the targets (*Figure 2—figure supplement 2*). We also note that our model further predicts that the magnitude of RPE to a reward predictive cue can be larger than the magnitude of RPE to the reward itself following random cues. This is because the predictive cues include the value of anticipation.

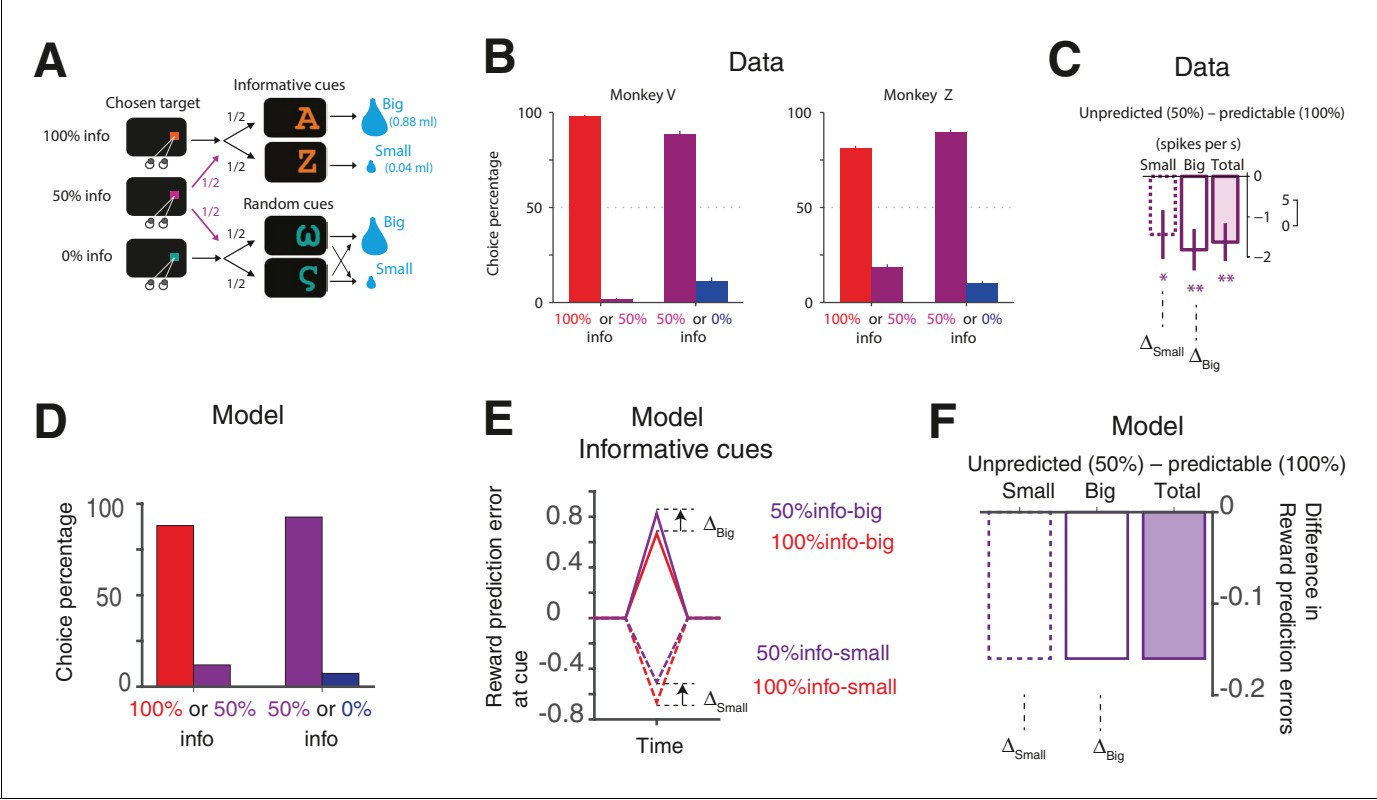

**Figure 2.** Our model accounts for the behavioral and neural findings in *Bromberg-Martin and Hikosaka (2011)*. (A) The task in (*Bromberg-Martin and Hikosaka, 2011*). On each trial, monkeys viewed a fixation point, used a saccadic eye movement to choose a colored visual target, viewed a visual cue, and received a big or small water reward. The three potential targets led to informative cues with 100%, 50% or 0% probability. (*Bromberg-Martin and Hikosaka, 2011*; reproduced with permission) (B) Monkeys strongly preferred to choose the target that led to a higher probability of viewing informative cues (*Bromberg-Martin and Hikosaka, 2011*; reproduced with permission). (C) The activity of lateral habenula neurons at the predicting cues following the 100% target (predictable) were different from the case where the cues followed the 50% target (unpredicted) (*Bromberg-Martin and Hikosaka, 2011*; reproduced with permission). The mean difference in firing rate between unpredicted and predictable cues are shown in case of small-reward and big-reward (the error bars indicate SEM.). (D) Our model predicts the preference for more informative targets. (E,F) Our model's RPE, which includes the anticipation of rewards, can account for the neural activity. Note the activity of the lateral habenula neurons is negatively correlated with RPE.

The following figure supplements are available for figure 2:

**Figure supplement 1.** Our model can capture the preference of info targets with a wide range of parameters.

**Figure supplement 2.** RPE-boosting of anticipation is necessary to capture the choice preference of monkeys reported in *Bromberg-Martin and Hikosaka (2011)*.

Our model also accounted for puzzling irrational gambling behaviors that have been reported in many experiments (*Spetch et al., 1990*; *Gipson et al., 2009*; *Eliaz and Schotter, 2010*; *Vasconcelos et al., 2015*). These include a perplexingly greater preference for a target offering 50% chance of reward over either a target offering 75% chance of reward, or a certain target offering 100% chance of reward, at least when the delays between the predicting cues and rewards are long (*Figure 3A–C*) (*Spetch et al., 1990*; *Gipson et al., 2009*).

*Figure 3D* shows that our model correctly predicted that the value of the 50% reward target (red line) would be smaller than that of the 100% reward target (blue line) when the delay $T_{delay}$ between the cues until rewards, and hence the contribution of anticipation, was small. Thus, the modelled subjects would prefer the reliable target (indicated by the blue background on left). However, when the delay $T_{delay}$ was increased, the contribution to the value coming from anticipation was boosted for the 50% reward target, but not for the certain target, because of the RPE associated with the

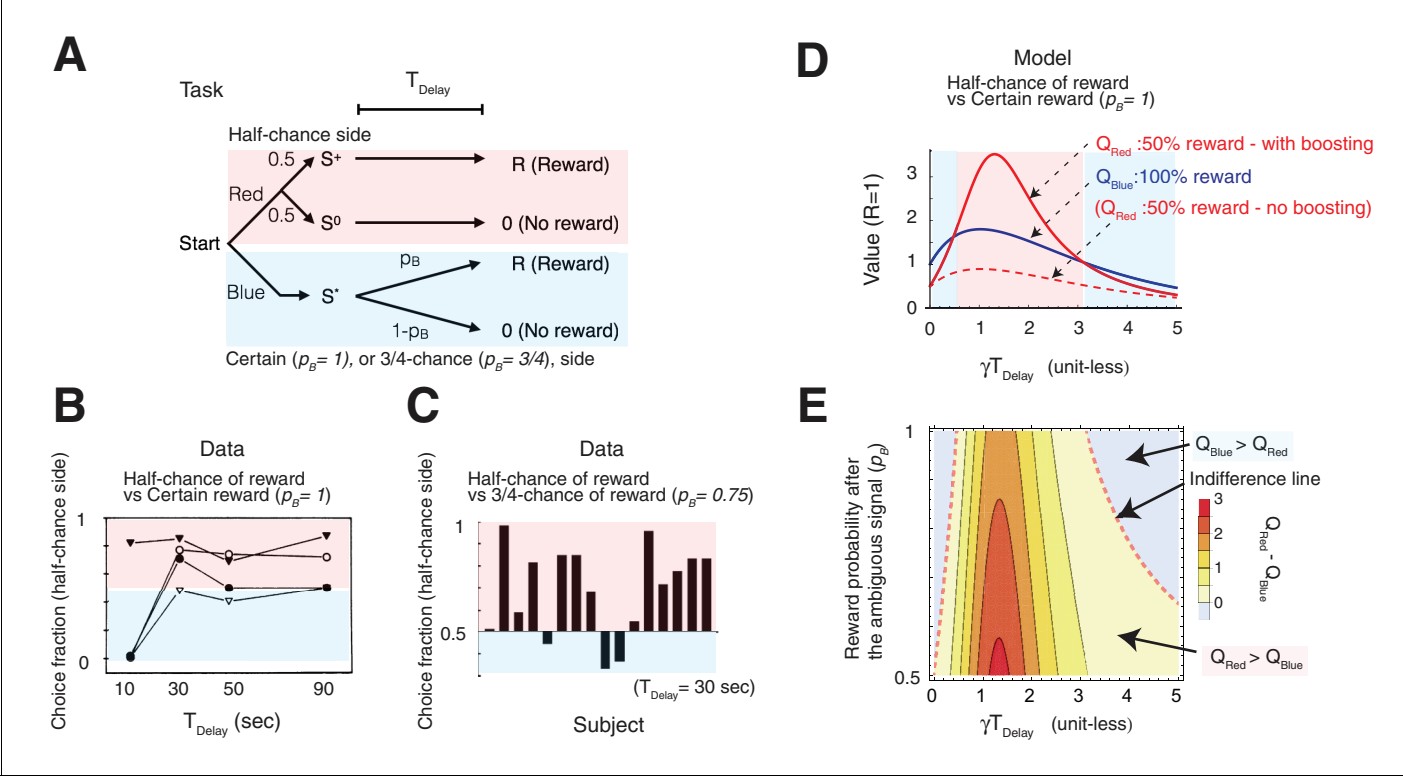

**Figure 3.** Our model accounts for a wide range of seemingly paradoxical findings of observing and information-seeking. (**A**) Abstraction of the pigeon tasks reported in *Spetch et al. (1990)*; *Gipson et al. 2009*). On each trial, subjects chose either of two colored targets (Red or Blue in this example). Given Red, cue $S^+$ or $S^0$ was presented, each with probability 0.5; was followed by a reward after time $T_{\text{Delay}}$, while $S^0$ was not followed by reward. Given Blue, a cue $S^*$ was presented, and reward possibly followed after the fixed time delay $T_{\text{Delay}}$ with probability $p_B$, or otherwise nothing. In *Spetch et al. (1990)*, $p_B=1$, and in *Gipson et al. (2009)* $p_B=0.75$. (**B**) Results with $p_B=1$ in *Spetch et al. (1990)*. Animals showed an increased preference for the less rewarding target (Red) as delay time $T_{\text{Delay}}$ was increased. The results of four animals are shown. (Adapted from *Spetch et al., 1990*) (**C**) Results with $p_B=0.75$ in *Gipson et al. (2009)*. Most animals preferred the informative but less rewarding target (Red). (Adapted from *Gipson et al., 2009*) (**D**) Our model predicted changes in the values of cues when $p_B=1$, accounting for (**B**). Thanks to the contribution of anticipation of rewards, both values first increase as the delay increased. Even though choosing Red provides fewer rewards, the prediction error boosts anticipation and hence the value of Red (solid red line), which eventually exceeds the value of Blue (solid blue line), given a suitably long delay. Without boosting, this does not happen (dotted red line). At the delay gets longer still, the values decay and the preference is reversed due to discounting. This second preference reversal is our model's novel prediction. Note that x-axis is unit-less and scaled by $\gamma$. (**E**) The changes in the values of Red and Blue targets across different probability conditions $p_B$. Our model predicted the reversal of preference across different probability conditions of $p_B$. The dotted red line represents when the target values were equal. We set parameters as $\nu/\gamma = 0.5, R = 1, \eta_0/\gamma = 3, c/\gamma = 3$.

The following figure supplements are available for figure 3:

**Figure supplement 1.** Related to *Figure 3*.

**Figure supplement 2.** Related to *Figure 3*.

former. This resulted in a preference for the lower probability target (indicated by the red background in the middle), which is consistent with the experimental finding shown in *Figure 3A,B* (*Spetch et al., 1990*). Note that RPE-boosting is again necessary to capture the reported suboptimal behaviors, as the expected non-boosted value of the 50% reward target would be always smaller than the value of sure reward target (see the dotted red line in *Figure 3D*). This is because in both cases, the rewards generate conventional anticipation.

Finally, as the time delay increased further, both values decayed, making the value of the certain target again greater than that of the 50% reward target (indicated by the blue background on right). This is because the discounting dominated the valuation, leaving the impact of anticipation relatively small. This second reversal is predicted by our model. However, it has not been observed

experimentally, other than in findings based on the use of hypothetical questionnaires in *Loewenstein (1987)*. (We note that one of the four animals in *Spetch et al. (1990)* did appear to show such a non monotonic preference; however, individual differences also appeared to be very large.)

The model can be used to interpolate between the experiments in *Spetch et al. (1990)* (*Figure 3B*) and *Gipson et al. (2009)* (*Figure 3C*), showing a full range of possible tradeoffs between the probability of reward and informative cueing. The phase diagram *Figure 3E* shows this trade-off for the task described in *Figure 3A*, as either the probability ($p_B$) of reward associated with the 0% info (blue) target, or the delay $T_{delay}$ change, while the reward probability of 100% info target (red) is fixed at 50%. The experiment in *Spetch et al. (1990)* (*Figure 3B*) corresponds to $p_B=1$, while the experiment in *Gipson et al. (2009)* (*Figure 3C*), which used a 75% chance target, corresponds to $p_B=0.75$. As we show in the case of $p_B \geq 0.5$, the chance of reward is higher for the 0% info (blue) target than the 100% info (red), 50% reward target in this diagram (except for the case $p_B=0.5$, where both targets offer 50% rewards). Hence the conventional reinforcement learning model would predict a preference for the 0% info (blue) target everywhere.

As seen in *Figure 3E*, the model predicted a similar reversal of preference as in *Figure 3D* across different probability conditions of $p_B>0.5$ We also confirmed that this prediction depends on neither the details of functional form by which RPE influenced anticipation, nor specific parameter values in the model (See Materials and methods and *Figure 3—figure supplement 1* ).

In these calculations, we set the value of no outcome to zero, implying a lack of dread in the no outcome condition. The reason for this was that there was no effect on the preference of pigeons when the delay between the cues signalling no-reward and the no-reward was changed (*Spetch et al., 1990*), while the impact of dread should change over the delay. Moreover changing the delay between choice and cues that signalled no-reward had no impact on preference (*McDevitt et al., 1997*). Note, however, that our results would still hold in case of adding a value to the no outcome. In fact, as detailed in the next section, we found in our human behavioral task that participants assigned the same magnitudes of values to reward and no-reward outcomes, and yet our model still accounted for the preference of advance information.

We note the generality of our model. It can account for other various experimental results in different conditions. This includes experiments showing the relative preference for a 100% info target with 25% chance of reward over a 50% info target with a 50% chance of reward (*Stagner and Zentall, 2010*; *Vasconcelos et al., 2015*), illustrated in *Figure 3—figure supplement 2*.

## Testing the preference for advance information about upcoming rewards across delays in human subjects

A consequence of our model is that the values of predictive cues will be affected by how long subjects subsequently have to wait for the reward – the dynamic changes in values across delay conditions shown in *Figure 1C*, *Figure 3D,E*. This has so far only been subject to rather limited tests in animal studies (*Spetch et al., 1990*). We therefore conducted a new human behavioral experiment to test these predictions.

In Experiment-1, 14 heterosexual male human volunteers chose between a 0% info target, which was followed by no cue (*Figure 4A*), and a 100% info target, which was immediately followed by cues that predicted the presence or absence of reward. The rewards were previously validated lascivious images of female models (*Crockett et al., 2013*). Using this type of primary rewards was crucial for our task design, as was also the case in *Crockett et al. (2013)*. This is because other types of rewards, such as monetary rewards, cannot be utilized by participants immediately on each trial as they become available.

Subjects experienced blocks of trials with fixed delays (2.5 s, 7.5 s, 20 s and 40 s), where the blocks were indicated by target colors. We set the chance of reward to $p_B=0.5$, consistent with the macaque experiments (*Bromberg-Martin and Hikosaka, 2009*; *2011*). Subjects were not told the exact reward probabilities, merely that rewards would be 'random'.

We confirmed our model's central prediction. Subjects showed increased preference for informative cues as the delay increased (the solid line in *Figure 4B* indicates group mean and SEM). Subjects were on average indifferent in the case of short delays (2.5 s and 7.5 s). However, they strongly preferred to choose the informative target in the case of longer delays (20 s, 40 s).

We fitted the choices to our reinforcement-learning (RL) model's trial-by-trial predictions, including the effects of learning from RPE-boosted anticipation. We used a form of hierarchical Bayesian

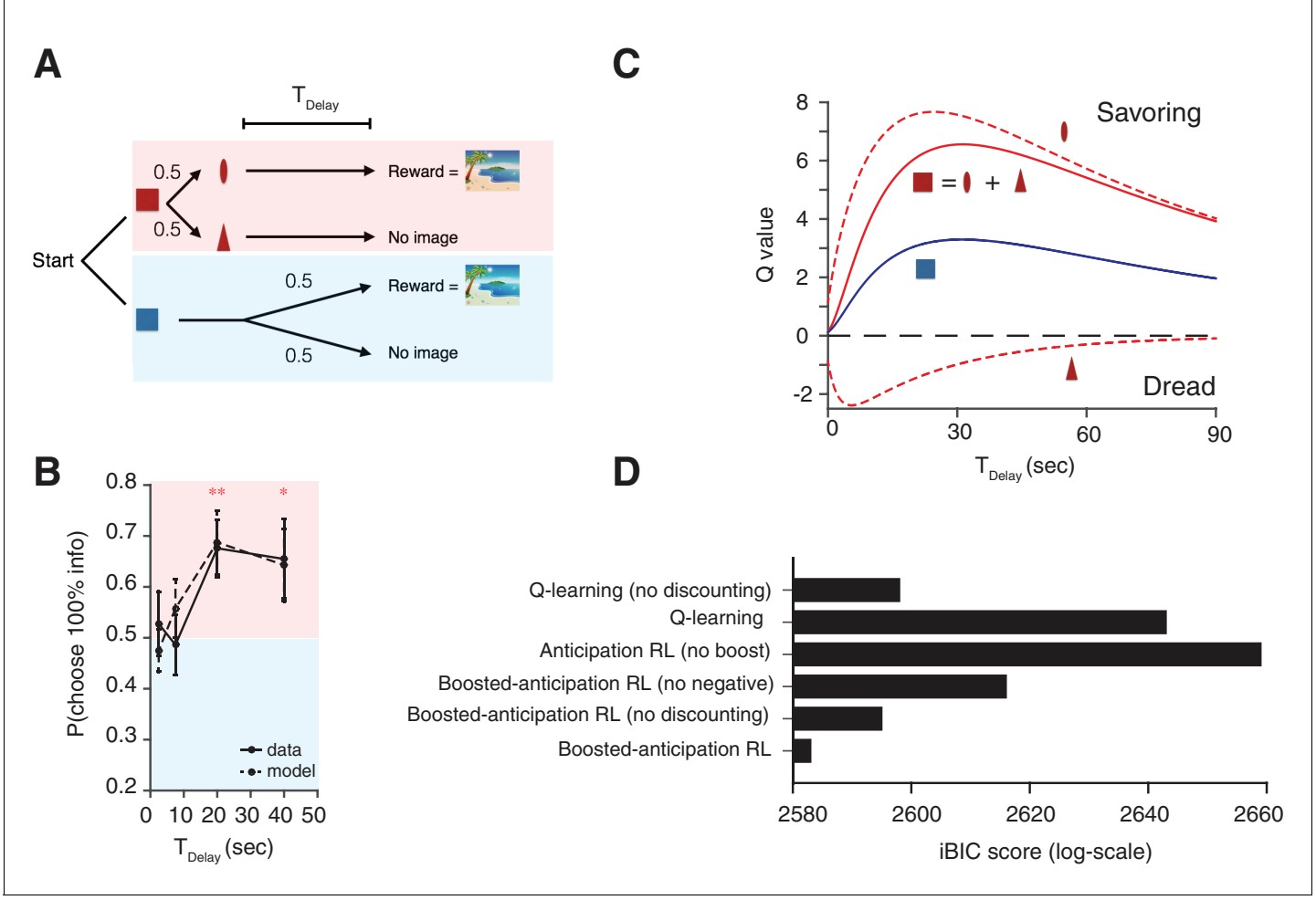

**Figure 4.** Human decision-making Experiment-1. (**A**) On each trial, subjects chose either of two colored targets (Red or Blue in this example). Given Red, cue $S^+$ (oval) or $S^0$ (triangle) was presented, each with probability 0.5; $S_+$; was followed by a reward (an erotic picture) after time $T_{\text{Delay}}$, while $S^0$ was not followed by reward. Given Blue, either a reward or nothing followed after the fixed time delay $T_{\text{Delay}}$ with probability 0.5 each. (**B**) Results. Human participants (n=14) showed a significant modulation of choice over delay conditions [one-way ANOVA, F(3,52)=3.09, p=0.035]. They showed a significant preference for the 100% info target (Red) for the case of long delays [20 s: $t(13) = 3.14$, $p = 0.0078$, 40 s: $t(13) = 2.60$, $p = 0.022$]. The mean +/- SEM indicated by the solid line. The dotted line shows simulated data using the fitted parameters. (**C**) Mean Q-values of targets and predicting cues estimated by the model. The value of informative cue is the mean of the reward predictive cue (oval), which has an inverted U-shape due to positive anticipation, and the no-reward predictive cue (triangle), which has the opposite U-shape due to negative anticipation. The positive anticipation peaks at around 25 s, which is consistent with animal studies shown in *Figure 3(B,C)*. See *Table 2* for the estimated model parameters. (**D**) Model comparison based on integrated Bayesian Information Criterion (iBIC) scores. The lower the score, the more favorable the model. Our model of RPE-boosted anticipation with a negative value for no-outcome enjoys significantly better score than the one without a negative value, the one without RPE-boosting, the one without temporal discounting, or other conventional Q-learning models with or without discounting.

The following figure supplements are available for figure 4:

**Figure supplement 1.** (**A**) Control experiment, where the first block and the last (5th) block of the experiment had the same delay duration of 2.5 s.

**Figure supplement 2.** The generated choice by the model without the negative value assigned to the no-reward outcome.

analysis to estimate group level parameters (*Huys et al., 2011*) (see Materials and methods for more details). We found that preferences generated from the estimated parameters were consistent with subjects' choices, as indicated by the black dotted line with the predicted standard error in *Figure 4B*. Our model predicted striking U-shape changes in the values of the 100% info target and of the 0% info target with respect to the changes in delay length (*Figure 4C*). The model enjoyed a

**Table 1.** iBIC scores. Related to *Figure 4*.

| Model | N of parameters | Parameters | iBIC |
|---|---|---|---|
| Q-learning (with no discounting) | 3 | $\alpha, R^+, R^-$ | 2598 |
| Q-learning (with discounting) | 5 | $\alpha, R^+, R^-, \gamma_+, \gamma_-$ | 2643 |
| Anticipation RL without RPE-boosting | 7 | $\alpha, R^+, R^-, \gamma_+ (= \gamma_-), \nu_+, \nu_-, \eta_0$ | 2659 |
| Boosted anticipation RL without $R^-$ | 4 | $\alpha, R^+, \gamma_+, \nu_+$ | 2616 |
| Boosted anticipation RL with no discounting | 5 | $\alpha, R^+, R^-, \nu_+, \nu_-$ | 2595 |
| Boosted anticipation RL | 6 | $\alpha, R^+, R^-, \gamma_+ (= \gamma_-), \nu_+, \nu_-$ | 2583 |

substantial iBIC score advantage over other possible well-studied reinforcement learning models, or our anticipation RL model without RPE-boosting (*Figure 4D* and *Table 1*).

To investigate possible adaptation to delay, we also ran a control experiment on an additional 11 subjects for whom we changed the order of the delays, and also repeated the same 2.5 s delay in the first and last (fifth) blocks. Preferences did not differ (*Figure 4—figure supplement 1A*) from beginning to end, suggesting stability. However, there was a moderately significant evidence of adaptation, with the effect of the 40 s delay being much greater following the extensive experience of 2.5 s delay than the 20 s delay (*Figure 4—figure supplement 1B*).

To investigate the robustness of the delay dependent preference of advance information further, we conducted an additional experiment (Experiment-2) on a newly recruited population of 31 participants (*Figure 5*). At the beginning of each trial, explicit cues provided participants with full information about the current delay condition (either 1 s, 5 s, 10 s, 20 s or 40 s) and the (constant, 50%) reward probability (*Figure 5A,B*). The basic structure of the task was similar to Experiment-1 (*Figure 5C*); participants had to choose either 100% info target or 0% info target. Crucially, though, the delay condition was randomized across trials. As seen in *Figure 5D* participants showed significant preference of the 100% info target when delay was long, replicating the results of Experiment-1. We also found no effect of the delay condition of a previous trial, providing further support for the absence of a block order confound in Experiment 1. (We computed each participant's preference of 100% info target at a particular delay condition $T_1$, conditioned on a previous delay condition $T_2$. We tested the effect of previous delay conditions $T_2$ over participants via one-way ANOVA for (1) the preference at $T_1$=1 s [$F(4, 111) = 2.36; p = 0.06$]; (2) the preference at $T_1$=5 s [$F(4, 101) = 0.85; p = 0.50$]; (3) the preference at $T_1$=10 s [$F(4, 99) = 1.45; p = 0.22$]; (4) the preference at $T_1$=20 s [$F(4, 87) = 1.48; p = 0.21$]; (5) the preference at $T_1$=40 s [$F(4, 98) = 0.75; \ p = 0.56$]). Also, Experiment-2 was designed to have an equal number of trials per delay condition, while Experiment-1 was designed to equalize the amount of time that participants spent in each condition (see Task Procedures in Materials and methods). The fact that we obtained the same results in both experiments illustrates the robustness of our findings.

In fitting the model, we were surprised to find that subjects assigned negative values to no-reward outcomes and its predicting cue (the bottom dotted line in *Figure 4C*, see also *Table 2*) for the estimated model parameters). This negativity *emerged* through our fitting, as we did not assume the sign of the value. Forcing this outcome to be worth 0 led to a significantly worse iBIC score (*Figure 4D* and *Table 1*). We found a particular effect of dread at the delay of 7.5 s, which the model interpreted as implying that the time scale of savoring was longer than that of dread (see $\nu^+$ and $\nu^-$ in *Table 2*). Omitting the negative value of the no-reward cue led to a failure to fit this effect (*Figure 4—figure supplement 2*).

Thus analysis using our model showed that the values of targets were computed via a competition between savoring and dread (*Figure 4C*). That is, when participants chose 0% info target, they experienced a mixture of the baseline savouring of possible reward and the baseline dread of possible no-reward during the wait period. On the other hand, when they chose the 100% info target, they experienced either savoring that was boosted by the RPE from the reward predictive cue, or dread that was boosted by the RPE from the no-reward predictive cue. It is this that required us to use the absolute value of the RPE to boost the effects of both savouring and dread. Specifically, RPE from the no-reward predictive cue also boosted the impact of dread, rather than damped it.

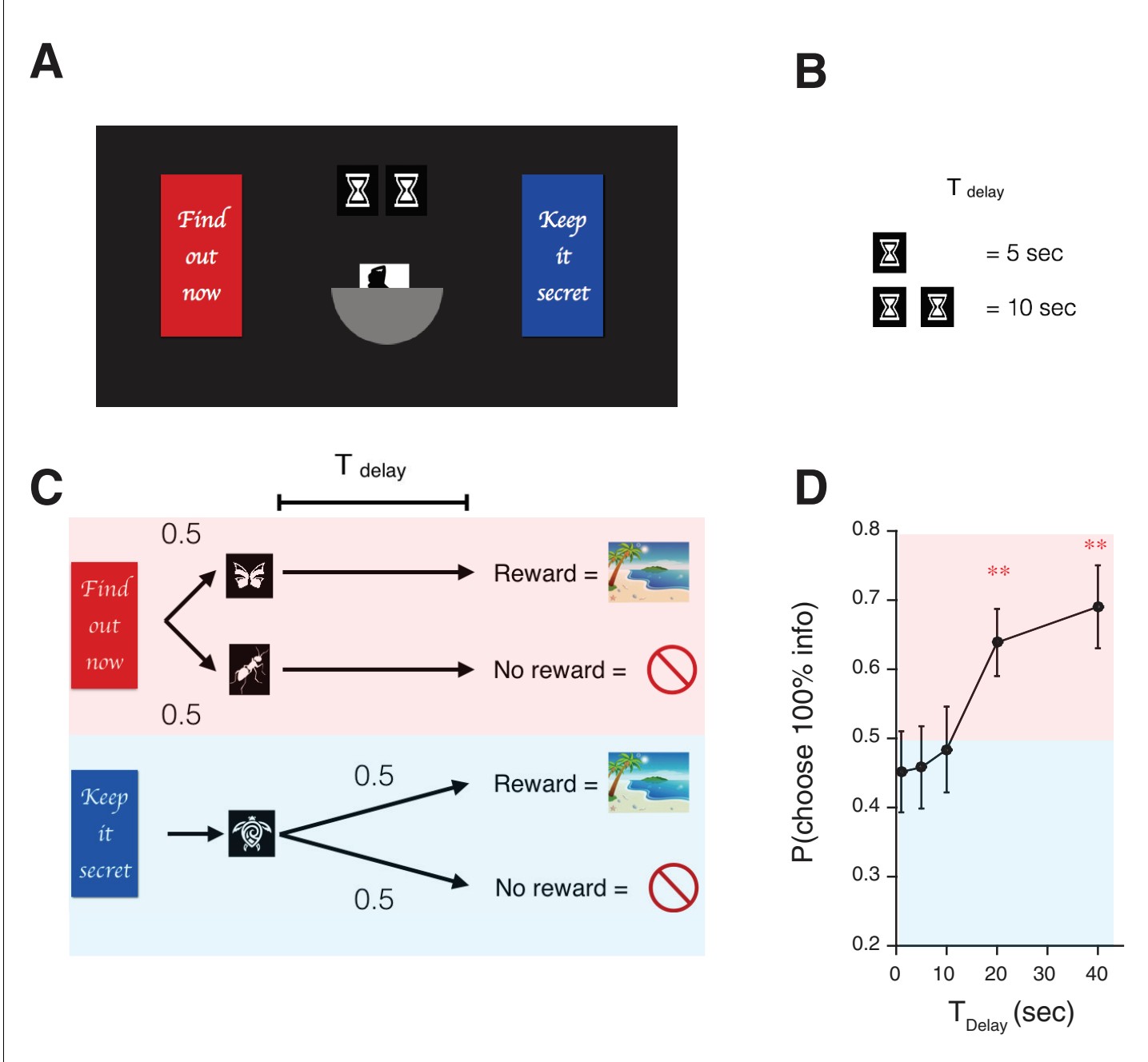

**Figure 5.** Human decision-making Experiment-2. (**A**) A screen-shot from the beginning of each trial. The meaning of targets ('Find out now' or 'Keep it secret'), the duration of $T_{\text{delay}}$ (the number of hourglass), and the chance of rewards (the hemisphere = 0.5) were indicated explicitly. (**B**) The number of hourglasses indicated the duration of $T_{\text{delay}}$ until reward. One hourglass indicated 5 s of $T_{\text{delay}}$. When $T_{\text{delay}} = 1$ s, a fraction of an hourglass was shown. This was instructed before the experiment began. The delay condition $T_{\text{delay}}$ was changed randomly across trials. (**C**) The task structure. The task structure was similar to Experiment-1, except that the 0% info target (Blue) was followed by a no-info cue, and an image symbolizing the lack of reward was presented when no reward outcome was delivered. (**D**) Results. Human participants (n=31) showed a significant modulation of choice over delay conditions [one-way ANOVA, F(4,150)=3.72, p=0.0065]. The choice fraction was not different from 0.5 at short delays [1 s: $t(30) = 0.83$, $p = 0.42$ 5 s: $t(30) = 0.70$, $p = 0.49$, 10 s: $t(30) = 0.26$, $p = 0.80$] but it was significantly different from 0.5 at long delays [20 s: $t(30) = 2.86$, $p = 0.0077$, 40 s: $t(30) = 3.17$, $p = 0.0035$], confirming our model's key prediction. The mean and +/- SEM are indicated by the point and error bar.

**Table 2.** Related to *Figure 4*. The group means $\mu$ that estimated by hierarchical Bayesian analysis for our human experiment.

| $\alpha$ | $cR^+$ | $cR^-$ | $\gamma_+ (= \gamma_-)$ | $\nu_+$ | $\nu_-$ |
|---|---|---|---|---|---|
| 0.17 | 0.85 | -0.84 | 0.041 ($sec^{-1}$) | 0.082 ($sec^{-1}$) | 0.41 ($sec^{-1}$) |

Note that the difference between the data and the RPE-boosted anticipation model without dread, seen in (*Figure 4—figure supplement 2*) at short delay periods, implies that the increase of preference of 100% info target was lower than what the model with only savouring can predict. This is because the preference of targets reflected the competition between positive savouring and negative dread, both of which changed non-monotonically with delay (*Figure 4C*).

More precisely, for delays wherein the impact of savoring and dread were similarly strong, choice preference remained at around the chance level. This phenomenon was confirmed in our Experiment-1 and Experiment-2 at short delay conditions ($<10$ s), where the choice probability remained around 0.5 (*Figure 4C*,*5D* and *Figure 4—figure supplement 2*). Since the timescale of dread was smaller than that for savoring, the effect of competition was present only at the short delay conditions. By contrast, at longer delay conditions ($>20$ s), dread was discounted and savoring became dominant. This resulted in a large increase of preference for the 100% info target.

This sudden increase in choice preference was caused by the non-monotonic Q-value functions of targets (*Figure 4C*). Our model comparison analysis also supported this conclusion (*Figure 4D*), where the model with non-monotonic value functions (our original model with temporal discounting with the estimated group mean of the discounting rate: $\gamma = 0.04\,s^{-1}$) outperformed the model with monotonic value functions (our model without temporal discounting $\gamma = 0$). The difference in the iBIC score was 12.

Note that temporal discounting of savoring at extremely long delays should also make subjects indifferent between 100% and 0% info targets. Unfortunately, we failed to confirm this prediction in the current study. In fact, analysis of our model suggests that in order to confirm this effect the delays concerned would need to be more than 135 s to enable detection of a difference between the preference at the extremely long delay and the preference at $T_{\text{Delay}} = 20$ s ($p = 0.05$, $n = 30$). Thus, we were not able to confirm this indifference in our task, and leave this for future studies. Nonetheless, we note that our model with temporal discounting outperformed a model without temporal discounting in terms of the iBIC scores (*Figure 4D*), where the latter model predicts monotonic value functions.

The magnitude of our discount rate (0.04 $s^{-1}$) is comparable with other intertemporal choice studies (e.g. 0.095 $s^{-1}$ for monetary rewards (*Schweighofer et al., 2008*), and 0.014 $s^{-1}$ for primary (juice) rewards (*McClure et al., 2007*). Note the latter was inferred using a double exponential model, with the faster decay being 0.014 $s^{-1}$ and the slower decay not being significantly different from 0 $s^{-1}$. Nonetheless, we should also point out that comparing discounting rates across different experimental designs is extremely difficult. For instance, the nature of discounting of primary rewards (e.g. juice or pleasant images) is very likely to be different from discounting of monetary rewards, as money cannot be spent immediately. In fact, *Reuben et al. (2010)* reported that primary rewards (chocolates) were more rapidly discounted than monetary rewards. It is also known that addicts discount the addictive substance at a higher rate than money (e.g. see (*Bickel et al., 1999*) for cigarettes, (*Bickel et al., 2011*) for cocaine). The characteristic timescales of these experiments were much longer (weeks or months) and discounting rates in these literatures are, however, very small compared to ours. This also suggests that comparisons across experiments could be very misleading, since discounting can be adaptive to experimental timescales (*Kable and Glimcher, 2010*).

We acknowledge that in the current study we did not test our model's prediction of preference reversal, as we designed the task so that the average amount of reward obtained from each target per trial was the same for both targets. This is a limitation of our current study, and we leave this issue to future investigations.

## Discussion

Although reward prediction errors (RPEs) have historically been treated as straightforward learning signals, there is increasing evidence that they also play more direct roles, including in classical conditioning (or Pavlovian) approach (*McClure et al., 2003*) and subjective well-being (*Rutledge et al., 2014*, *2015*). The latter study found that subjective well-being, or happiness, is influenced more prominently by RPE than reward itself, which has echoes with an older idea often referred to as the 'hedonic treadmill' (*Brickman and Campbell, 1971*; *Frederick and Loewenstein, 1999*). Here we considered a further contribution of RPEs, stemming from their ability to boost the value of the anticipation of reward.

RPE-boosted anticipation provides a natural account for gambling behaviors. Indeed, our model further predicts that the tendency to be risk-seeking or risk-averse is subject to change as a function of the delay between the cues and rewards. This has important consequences for gambling, as well as the nature and measurement of risk attitudes in general. Specifically, our findings suggest that an unexpected prize will have greater motivational impact when there is a moderate delay between its revelation and its realization. Further experiments will be required to confirm this novel prediction in the context of more conventional economic gambling tasks. We also note that it has been shown in macaque monkeys that changing inter-trial-intervals can impact risk sensitivity (*McCoy and Platt, 2005*; *Hayden and Platt, 2007*). This indicates that the anticipation and discounting of future outcomes over multiple trials may also play important roles in determining risk attitudes.

RPE-boosted anticipation, like many apparently Pavlovian behaviors that involve innate responses, appears evidently suboptimal. As we have seen, choice can be strikingly non-normative. Our results are consistent with notions such as curiosity/exploration bonuses, and uncertainty aversion (*Loewenstein, 1994*; *Caplin and Leahy, 2001*; *Litman, 2005*; *Daw et al., 2006*; *Fiorillo, 2011*; *Friston et al., 2013*; *2015*; *Gottlieb et al., 2013*; *Blanchard et al., 2015a*; *Kidd and Hayden, 2015*). However, whether the behaviors reflect mechanistic constraints on neural computation (*Kakade and Dayan, 2002*), or a suitable adaptation to typical evolutionary environments, remains a question for further research.

In our human experiments, we found that participants assigned a negative value to a no-outcome. This appears not to be the case in reported pigeon experiments. One idea is that this negative value emerges from a form of normalization of subjective values (*Tobler et al., 2005*; *2007*; *Louie et al., 2013*) consistent with the finding that human subjects can assign 'unpleasantness' to no-reward stimuli (*Tobler et al., 2007*). The effect in our task would be that subjects would apparently experience the anticipation of both positive and negative outcomes in this task as being pleasant for the reward predictive cue (savouring) but unpleasant for the no-outcome predictive cue (dread). This was confirmed in informal debriefing after the experiment. Note that in the monkey experiments (*Bromberg-Martin and Hikosaka, 2009*; *2011*), the lower-value outcome still involved an actual reward, albeit of a smaller size.

We showed that responses of habenula neurons to reward predictive cues, which have been proposed as an 'information prediction error' (*Bromberg-Martin and Hikosaka, 2009*; *2011*), could be accounted for by our model in terms of conventional reward prediction errors. This is because our model included the value of anticipation of rewards that can be boosted by RPE.

Further studies are necessary to explore the calculation and representation of the anticipation itself, for both savouring (of positive outcomes) and dread (of negative ones). We note recent experimental findings in basal forebrain suggest seductive similarities (*Monosov and Hikosaka, 2013*; *Monosov et al., 2015*), while other brain areas, such as ventral striatum (*Jensen et al., 2003*; *Hariri et al., 2006*; *Salimpoor et al., 2011*), posterior insula and anterior cingulate cortex (*Berns et al., 2006*; *Blanchard et al., 2015b*), may also contribute. Furthermore, ramping dopamine signals toward the delivery of rewards (if they generally exist, see [*Morris et al., 2004*]) could also be related to the anticipation of rewards (*Fiorillo et al., 2003*; *Howe et al., 2013*; *Lloyd and Dayan, 2015*; *Hamid et al., 2016*; *Hart et al., 2015*), while dopamine neurons have also been shown to manifest a stronger phasic response to one predicting an uncertain reward than to a cue predicting a certain reward (*Fiorillo, 2011*). It would also be interesting to study the difference between savouring and dread, particularly given debates concerning the encoding of information about punishment *Fiorillo et al. (2003)*, *Brischoux et al. (2009)*, *Lammel et al. (2014)*, and about the symmetry or

otherwise between the encoding of positive and negative prediction errors for reward (*Fiorillo, 2013*; *Hart et al., 2014*)

One issue that merits further future study is adaptation to different delays. It has been shown that human subjects are capable of optimizing their discounting rate according to task demands (*Schweighofer et al., 2006*), and also that the discounting may be computed relative to the timing of other available options, rather than absolute time (*Kable and Glimcher, 2010*). Our control experiment for Experiment-1 showed a signature of adaptation (*Figure 4—figure supplement 1A*), with subjects reacting differently to a sudden large increase in delays after many trials with small delays, compared to a gradual increase in delays as in Experiment-1. However we found no evidence for this effect in our Experiment-2 in which delay conditions were randomized on a trial-by-trial basis. It would be interesting to study this further, in relation to the uncertainty in timing of rewards. In our task there was no effect of timing uncertainty on choice, as both targets are associated with the same delay. However it would become important if a task involves a choice between targets with different delay conditions. Furthermore, if the prediction error could influence subjective time (for instance via a known effect of dopamine on aspects of timing [*Gibbon et al., 1997*]), then this could have complex additional effects on anticipation.

In sum, we account for a well-described preference for observing behavior through a suggestion that reward prediction errors modulate the contribution to subjective value that arises from the anticipation of upcoming rewards. Our study provides a new perspective on reward-based learning and decision-making under uncertainty, and may be of special relevance to gambling and addiction.

## Materials and methods

### Participants
56 heterosexual male participants (age 18–40) were recruited from the UCL community. Participants were paid 10 British pounds at the end of the experiment. Participants provided informed consent for their participation in the study, which was approved by the UCL ethics committee (UCL Research Ethics Reference: 3450/002).

### Task procedures
#### Experiment-1
14 participants performed the main experimental task, which we call Experiment-1. First, the procedure was explained to them; they then underwent a practice session consisting of 30 trials with the same format as the main session except with different target colors and reward images. The practice session was followed by the experimental trials, involving four blocks with different (not-randomized) delay conditions: 2.5 s (90 trials), 7.5 s (36 trials), 20 s (18 trials), 40 s (18 trials).

At the beginning of each trial, two rectangular targets with different colors appeared side-by-side on the screen. The targets were either informative or uninformative, indicated by colors. Participants were instructed that one of the colors led to 'signs' that indicate the future presence of a reward or no-reward (see below). The positions of the targets were determined randomly on each trial. Participants had to choose the left or right target by pressing 'F' or 'J' respectively within 3 s; responses on other keys or that were longer than 3 s resulted in a penalty of 3 s, followed by another trial. Once a target had been chosen, it remained stationary on the screen for 1.5 s; the other target having been extinguished. If participants chose the informative target, then one of the two shape cues (triangle or oval) with the same color as the target randomly appeared in the center of the screen. Each shape consistently indicated that the outcome would be reward or no-reward. The cue remained visible for the length of the delay determined by the block. If participants chose the no-information target, the target disappeared after 1.5 s, and no cue appeared on the screen for the delay period, followed randomly by a reward or no-reward. In case of a reward, an image of an attractive female model was presented for 1.3 s. In case of no-reward, no image was shown for this period. After the reward or no-reward period, a blank screen was shown for 1.2 s before the next trial began, when two targets appeared on the screen.

## Experiment-1 - control

A control experiment was conducted on additional 11 participants. Everything was the same as for the main task, except that these participants faced the delay conditions in a different order from the main task: 2.5 s (90 trials), 40 s (18 trials), 20 s (18 trials), 7.5 s (36 trials), and then an additional block with the same delay as the first block 2.5 s (36 trials).

As seen in *Figure 4—figure supplement 1A*, subjects' choices in the first and fifth blocks (under the same delay condition; 2.5 s) did not differ. We found, however, the change between the first and the second blocks was so dramatic in terms of the change in $T_{delay}$ that subjects behaved differently between the first (2.5 s) and second (40 s) block compared to the main task, in which the delay length increased gradually across the blocks *Figure 4—figure supplement 1B*. It is possible that the perception of delay was influenced by history; something that we did not take into the account in our model. Hence we focused on our analysis on the main task subjects.

## Experiment-2

An additional 31 participants performed Experiment-2. The basic structure was similar to Experiment-1, however, in this task, 1) the delay conditions $T_{\text{Delay}}$ were randomized trial-by-trial; 2) the delay length (1 s, 5 s, 10 s, 20 s or 40 s) was explicitly indicated by a cue at the beginning of each trial; 3) the targets, either informative or uninformative, were explicitly indicated by messages presented at targets; 4) the uninformative target was also followed by a cue, rather than a blank screen; 5) a no-reward outcome was indicated by a no-entry sign.

As in Experiment-1, the procedure was explained to participants before they underwent a practice session consisting of 10 trials (2 trials per each delay condition).

At the beginning of each trial, pictures of hourglasses and a silhouette of woman covered by a semicircle appeared on the screen. Participants were instructed that the number of hourglasses indicated the delay between their choice and the reward delivery ($T_{\text{Delay}}$), and that one hourglass indicated 5 s of delay. When the delay was 1 s, a fraction of an hourglass was displayed on the screen; this was also instructed to participants. Further, they were instructed that the semicircle appeared on the screen indicating a 50% chance of getting a reward. The delay conditions were randomized across trials.

After 500 ms, two rectangular targets with different colors appeared side by side on the screen. One target always presented the message 'Find out now', while the other target had a message 'Keep it Secret'. The same message appeared on the same colored target across trials, but the sides on which target and message appeared were randomized across trials. Participants had to choose the left or right target by pressing 'F' or 'J'. Once a target had been chosen, the chosen target was highlighted by yellow exterior for 1.5 s. If participants chose the informative target, then one of the two cues (a symbolic picture of a butterfly or an ant) randomly appeared at the center of the screen. Each cue consistently indicated that the outcome would be reward or no-reward. The cue remained visible for the length of the delay that was determined by the number of hourglasses presented at the beginning of each trial. If participants chose the no-information target, a cue (a symbolic picture of a turtle) appeared on the screen for the delay period, followed randomly by a reward or no-reward. In case of a reward, an image of an attractive female model was presented for 1.3 s. In case of no-reward, the image of a no-entry sign was presented for 1.3 s. After the reward or no-reward period, a blank screen was shown for 1.2 s before the next trial began. In the main task, each participant performed 25 trials (5 trials per delay condition), and received 10 British pounds at the end of experiment.

### Reward images

We sought to use basic rewards that could be consumed by subjects on each trial at the time of provision. We therefore employed images of female models that had previously been rated by heterosexual male subjects (*Crockett et al., 2013*). In case of reward, a random one of the top 100 highest rated images was presented to subjects without replacement.

### Analysis

We sought to determine the distribution of model parameters $\mathbf{h}$. Thus following (*Huys et al., 2011*), we conducted a hierarchical Bayesian, random effects analysis, where the (suitably transformed)

parameters $\mathbf{h}_i$ of individual $i$ are treated as a random sample from a population distribution, which we assume to be Gaussian, with means and variance $\boldsymbol{\theta} = \{\boldsymbol{\mu}_\theta, \boldsymbol{\Sigma}_\theta\}$.

The prior group distribution $\boldsymbol{\theta}$ can be set as the maximum likelihood estimate:

$$\boldsymbol{\theta}^{ML} \approx \mathrm{argmax}_{\boldsymbol{\theta}}\{p(D|\boldsymbol{\theta})\}$$

$$= \mathrm{argmax}_{\boldsymbol{\theta}}\left\{\prod_{i=1}^{N}\int d\mathbf{h_i}\, p(D_i|\mathbf{h}_i)p(\mathbf{h}_i|\boldsymbol{\theta})\right\} \tag{3}$$

We optimized $\boldsymbol{\theta}$ using an approximate Expectation-Maximization procedure. For the E-step of the k-th iteration, we employed a Laplace approximation, obtaining,

$$\mathbf{m}_i^k \approx \mathrm{argmax}_{\mathbf{h}}\left\{p(D_i|\mathbf{h})p(\mathbf{h}|\boldsymbol{\theta}^{k-1})\right\} \tag{4}$$

$$p(\mathbf{h}_i^k|D_i) \approx \mathcal{N}\left(\mathbf{m}_i^k, \boldsymbol{\Sigma}_i^k\right), \tag{5}$$

where $\mathcal{N}\left(\mathbf{m}_i^k, \boldsymbol{\Sigma}_i^k\right)$ is the Normal distribution with the mean $\mathbf{m}_i^k$ and the covariance $\boldsymbol{\Sigma}_i^k$ that is obtained from the inverse Hessian around $\mathbf{m}_i^k$. For the M step:

$$\boldsymbol{\mu}_\theta^{k+1} = \frac{1}{N}\sum_{i=1}^{N}\mathbf{m}_i^k \tag{6}$$

$$\boldsymbol{\Sigma}_\theta^{k+1} = \frac{1}{N}\sum_{i=1}^{N}\left(\mathbf{m}_i^k\mathbf{m}_i^{k\mathbf{T}} + \boldsymbol{\Sigma}_i^k\right) - \boldsymbol{\mu}_\theta^{k+1}\boldsymbol{\mu}_\theta^{k+1\mathbf{T}}. \tag{7}$$

For simplicity, we assumed that the covariance $\Sigma_\theta^k$ had zero off-diagonal terms, assuming that the effects were independent. Also, in order to treat different delay conditions equally, we randomly sub-sampled the trials to equalize the number used per condition in order to calculate the statistics. Additionally we obtained the same results by normalizing the posterior for each delay condition when estimating the expectation.

For the model-free data analysis, we used the t-test, as the data passed the Shapiro-Wilk normality test and the paired F-test for equal variances for each and between conditions.

## Model comparison

We compared models according to their integrated Bayes Information Criterion (iBIC) scores, based on a flat prior over models. We analysed model log likelihood $\log p(D|M)$:

$$\log p(D|M) = \int d\theta p(D|\theta)p(\theta|M) \tag{8}$$

$$\approx -\frac{1}{2}\mathrm{iBIC} = \log p\left(D|\theta^{ML}\right) - \frac{1}{2}|M|\log|D|, \tag{9}$$

where iBIC is the *integrated* Baysian Information Criterion, $|M|$ is the number of fitted parameters of the prior and $|D|$ is the number of data points (total number of choices made by all subjects). Here, $\log p(D|\theta^{ML})$ can be computed by integrating out individual parameters:

$$\log p\left(D|\theta^{ML}\right) = \sum_i \log\int d\mathbf{h}p(D_i|\mathbf{h})p\left(\mathbf{h}|\theta^{ML}\right) \tag{10}$$

$$\approx \sum_i \log\frac{1}{K}\sum_{j=1}^{K}p\left(D_i|\mathbf{h}^j\right), \tag{11}$$

where we approximated the integral as the average over $K$ samples $\mathbf{h}^j$'s generated from the prior $p(\mathbf{h}|\theta^{ML})$.

As seen in *Table 1*, our model of anticipation fit better than conventional Q-learning models with or without discounting (the latter two models being equivalent to our model with parameters set such that there is no anticipation.)

## Model

We describe our model for the case of a simple conditioning task. Suppose that a subject takes an action and receives a reward predictive cue $S^+$ at $t = 0$ with a probability of $q$ followed by a reward $R$ at $t = T(= T_{\mathrm{Delay}})$, or no-reward predictive cue $S^0$ at $t = 0$ with a probability of $1 - q$ followed by no reward. Following (**Loewenstein, 1987**; **Berns et al., 2006**; **Story et al., 2013**), the anticipation of the reward at time $t$ is worth $a(t) = Re^{-\nu(T-t)}$, where $\nu$ governs its rate. Including $R$ itself, and taking temporal discounting into account, the total value of the reward predictive cue, $Q_{S^+}$, is

$$
\begin{aligned}
Q_{S^+} &= \eta V^{[\mathrm{Anticipation}]} + V^{[\mathrm{Reward}]} \\
&= \eta \int_0^T e^{-\gamma t'} a(t') dt' + Re^{-\gamma T} \\
&= \eta \frac{R}{\nu - \gamma} \left( e^{-\gamma T} - e^{-\nu T} \right) + Re^{-\gamma T},
\end{aligned}
\tag{12}
$$

where $\eta$ is the relative weight of anticipation and $\gamma$ is the discounting rate. In previous work, $\eta$ has been treated as a constant; however, here we propose that it can vary with the prediction error $\delta_{pe}$ at the predicting cue. While the simplest form is the linear relationship given by **Equation (2)**, our model's behavior does not depend on the details of the RPE dependence of anticipation. In fact, one can instead assume

$$
\eta = \eta_0 + c_1 \tanh\left( c_2 |\delta_{pe}| \right)
\tag{13}
$$

where $c_1$ and $c_2$ are constants, or

$$
\eta = \eta_0 + c\theta\left( |\delta_{pe}| \right)
\tag{14}
$$

where $\theta(x)$ is the step function that we define: $\theta(x) = 1$ for $x > 0$ and $\theta(x) = 0$ for $x \leq 0$. All the findings in this paper hold for **Equation (2)**, **Equation (13)**, and **Equation (14)** (see **Figure 3—figure supplement 1**). In fact, **Equation (2)** and **Equation (14)** can be thought as different approximations to **Equation (13)** (see below).

Note that in our model, RPE affects the total value $Q_{S^+}$, which also affects subsequent RPEs. One might therefore wonder whether there is a stable value for the cue. In the Results, we introduced a linear ansatz for the boosting of anticipation on RPE (**Equation (2)**). In a wide range of parameter regime, this ansatz has a stable, self-consistent, solution; however, in a small parameter regime, the linear ansatz fails to provide such solutions. This is because the linear assumption allows *unbounded* boosting. Crudely, the RPE can usually be expressed as a linear combination of Q-values. Thus, the following equation has to have a real solution:

$$
\delta_{pe} = \alpha |\delta_{pe}| + \beta,
\tag{15}
$$

where $\alpha, \beta$ are determined by the task. The solution is the intercept of two lines: $y = \delta_{pe}$ and $y = \alpha |\delta_{pe}| + \beta$, which does not exist when $\alpha > 1$ and $\beta > 0$.

In this example, the prediction error at the reward predictive cue is positive, and

$$
\begin{aligned}
\delta_{pe} &= Q_{S^+} - qQ_{S^+} \\
&= (1 - q)Q_{S^+} \\
&= (1 - q)\left( \left( \eta_0 + c\delta_{pe} \right) V^{[\mathrm{Anticipation}]} + V^{[\mathrm{Reward}]} \right).
\end{aligned}
\tag{16}
$$

This has a solution at $\delta_{pe} > 0$ only if

$$
(1 - q)cV^{[\mathrm{Anticipation}]} < 1.
\tag{17}
$$

This condition can be violated, for instance, if $c$ is very large (more precisely, if $c > \frac{1-q}{V^{[\mathrm{Anticipation}]}}$). Roughly speaking, the stability condition is violated when the boosted anticipation is very large.

If it indeed exists, the solution is

$$\delta_{pe} \;=\; \frac{(1-q)\big(\eta_0 V^{[\text{Anticipation}]} + V^{[\text{Reward}]}\big)}{1-(1-q)cV^{[\text{Anticipation}]}} \tag{18}$$

which gives

$$Q_{S^+} \;=\; \frac{\eta_0 V^{[\text{Anticipation}]} + V^{[\text{Reward}]}}{1-(1-q)cV^{[\text{Anticipation}]}}. \tag{19}$$

To avoid the stability problem problem, one can instead assume *Equation (13)*. This is a more general form which leads to the self-consistency equation:

$$\delta_{pe} = \alpha \tanh\big(c_2 |\delta_{pe}|\big) + \beta, \tag{20}$$

which requires $y = \delta_{pe}$ and $y = \alpha \tanh\big(c_2|\delta_{pe}|\big) + \beta$ to intersect. This can happen for any real $\alpha, \beta$. However, importantly, our model's behavior does not depend on the details of the RPE dependence of anticipation (see *Figure 3—figure supplement 1* ). Hence we did not attempt to determine the exact functional form.

Note that *Equation (2)* can be thought of as an approximation to *Equation (13)* when $c_2|\delta_{pe}|$ is small. In the limit of $c_2 \to \infty$, on the other hand, *Equation (13)* becomes *Equation (14)*. *Equation (14)* can be used instead of *Equation (13)* when $c_2|\delta_{pe}|$ is large; or also can be used as an approximation of *Equation (2)* when the size of RPE is roughly the same from trial to trial.

## Model application 1: *Bromberg-Martin and Hikosaka task (2011)*

To see how the model works, take the task introduced in *Bromberg-Martin and Hikosaka (2011)* (*Figure 2A*). We assume that the 100% info target is followed randomly by a cue $S_{\text{Big}}$ that is always followed by a big reward $R_{\text{Big}}$ after a delay $T$, or by a cue $S_{\text{Small}}$ that is always followed by a small reward $R_{\text{Small}}(< R_{\text{Big}})$. The 0% info target is followed by a cue $S_{\text{Random}}$, which is followed by the reward $R_{\text{Big}}$ or $R_{\text{Small}}$ with equal probabilities. The 50% info target is followed by either of the three cues $S_{\text{Big}}$, $S_{\text{Small}}$, or $S_{\text{Random}}$ with a probability of $1/4$, $1/4$ or $1/2$, respectively.

The expected values of targets $(Q_{100\%}, Q_{50\%}, Q_{0\%})$ are

$$Q_{100\%} \;=\; \frac{Q_{S_{\text{Big}}} + Q_{S_{\text{Small}}}}{2} \tag{21}$$

$$Q_{0\%} \;=\; Q_{S_{\text{Random}}} \tag{22}$$

$$Q_{50\%} \;=\; \frac{Q_{S_{\text{Big}}} + Q_{S_{\text{Small}}}}{4} + \frac{Q_{S_{\text{Random}}}}{2} \tag{23}$$

where $Q_{S_{\text{Big}}}$, $Q_{S_{\text{Small}}}$, $Q_{S_{\text{Random}}}$ are the expected values of cues $S_{\text{Big}}$, $S_{\text{Small}}$, $S_{\text{Random}}$, respectively. The RPE at the cues depends on the chosen target, implying that the average values of the cues can be expressed (assuming that the transition probabilities are properly learned) as

$$Q_{S_{\text{Big}}} \;=\; \frac{V_{S_{\text{Big}},100\%} + V_{S_{\text{Big}},50\%}}{2} \tag{24}$$

$$Q_{S_{\text{Small}}} \;=\; \frac{V_{S_{\text{Small}},100\%} + V_{S_{\text{Small}},50\%}}{2} \tag{25}$$

$$Q_{S_{\text{Random}}} \;=\; \frac{V_{S_{\text{Random}},50\%} + V_{S_{\text{Random}},0\%}}{2} \tag{26}$$

where $V_{S_j,X\%}$ is the mean values of the cues $S_j$ ($j = $ Big, Small, or Random) in case following the $X\%$ info target ($X = 100, 50$, or $0$):

$$V_{S_{\text{Big}},X\%,} = \eta^{S_{\text{Big}},X\%} A_{\text{Big}} + B_{\text{Big}} \tag{27}$$

$$V_{S_{\text{Small}},X\%} = \eta^{S_{\text{Small}},X\%} A_{\text{Small}} + B_{\text{Small}} \tag{28}$$

$$V_{S_{\text{Random}},X\%,} = \frac{\eta^{S_{\text{Random}},X\%} A_{\text{Big}} + B_{\text{Big}} + \eta^{S_{\text{Random}},X\%} A_{\text{Small}} + B_{\text{Small}}}{2} \tag{29}$$

where, under the assumption *Equation (2)*,

$$\eta^{S_j,X\%} = \eta_0 + c \left| \delta_{pe}^{S_j,X\%} \right|, \tag{30}$$

where $\delta_{pe}^{S_j,X\%}$ is RPE at the cue $S_j$ after choosing $X\%$ target

$$\delta_{pe}^{S_j,X\%} = Q_{S_j} - Q_{X\%} \tag{31}$$

and $A_l$ and $B_l$ ($l$ is the reward size; in our case Big or Small) are the anticipation and the reward itself:

$$A_l = \frac{R_l}{\nu - \gamma} \left( e^{-\gamma T} - e^{-\nu T} \right) \tag{32}$$

$$B_l = R_l e^{-\gamma T} \tag{33}$$

Note that we ignored any anticipation of the cues themselves after the choice. This would not alter the qualitative predictions of the model. Recent experiments (*McDevitt et al., 1997*; *2016*) showed that delaying the timing of reward predictive cues decreased the preference for the informative target. This is consistent with our model because delaying the cue presentation means decreasing the wait time; hence leads to a smaller impact of boosted anticipation. In our experiment, the time between choice and the cue presentation was too short to be significant.

The probability of choosing $X\%$ target over $Y\%$ target, $P_{X\%-Y\%}$, is assumed to be a sigmoid function of the difference between the target values:

$$P_{X\%-Y\%} = \frac{1}{1 + e^{-\frac{Q_{X\%} - Q_{Y\%}}{\sigma}}} \tag{34}$$

These equations account well for the behavioral and neuronal findings in *Bromberg-Martin and Hikosaka (2011)*. The results in *Figure 2D,E* are obtained from these equations with $R_{\text{Big}} = 0.88, R_{\text{Small}} = 0.04, \nu = 0.5 \text{ sec}^{-1}, T_{\text{Delay}} = 2.25 \text{ sec}, \gamma = 0.1 \text{ sec}^{-1}, \sigma = 0.08$.

## Model application 2: Spetch et al. task (1990) (and also Gipson et al. task (2009), Stagner and Zentall (2010) task.)

*Spetch et al. (1990)* reported the striking finding that pigeons can prefer a target that is followed by reward with a probability of $0.5$ over a target that is always followed by a reward under certain conditions. Here we show that our model can also account for this surprisingly 'irrational' behavior. A generalized version of the task is schematically shown in *Figure 3A*. On each trial, a subject chooses either of two colored targets (Red or Blue in this example). If Red is chosen, one of the cues $S^+$ or $S^0$ is randomly presented, where $S^+$ is always followed by a reward after time $T_{\text{Delay}}$, while $S^0$ is never followed by reward. If Blue is chosen, a cue $S^*$ is presented and a reward is followed after the fixed time delay $T_{\text{Delay}}$ with a probability of $p_B$. The task in *Spetch et al. (1990)* corresponds to the case with $p_B = 1$, and the task in *Gipson et al. (2009)* corresponds to the case with $p_B$=0.75. In both cases, by always choosing Blue, animals can get the maximum amount of rewards; however, it has been shown that animals can prefer to choose Red over Blue (*Spetch et al., 1990*; *Gipson et al., 2009*), where the preference of Red with $p_B = 1$ appeared to be heavily dependent on the length of the delay between the predicting cues and the delivery of rewards.

Our model can account for the irrational behaviors. We first determine the value of choice, $Q_{\text{Red}}$. Under the linear ansatz (2), the prediction error at the cue $S^+$ is

$$\delta_{pe} = Q_{S^+} - Q_{\text{Red}} \tag{35}$$

$$= \frac{1}{2} Q_{S^+} \tag{36}$$

$$= \frac{1}{2} \big( (\eta_0 + c\delta_{pe})A + B \big), \tag{37}$$

or

$$\delta_{pe} = \frac{\frac{1}{2}(\eta_0 A + B)}{1 - \frac{1}{2}cA} \tag{38}$$

and

$$Q_{S^+} = \frac{(\eta_0 A + B)}{1 - \frac{1}{2}cA} \tag{39}$$

$$Q_{\text{red}} = \frac{Q_{S^+} + Q_{S^0}}{2} \tag{40}$$

$$= \frac{1}{2} \frac{(\eta_0 A + B)}{1 - \frac{1}{2}cA}, \tag{41}$$

where

$$A = \frac{R}{\nu - \gamma} \big( e^{-\gamma T} - e^{-\nu T} \big) \tag{42}$$

$$B = R e^{-\gamma T} \tag{43}$$

with $R$ being the size of reward. *Equation (39)* shows how the Q-value of reward predictive cue is boosted. When there is no boosting, $c = 0$, the denominator is 1. Increasing the boosting $c$ will decrease the denominator; hence it will increase the Q-value. Note that the denominator is assumed to be positive within the linear ansatz. The value of choice Blue is simply

$$Q_{\text{Blue}} = p_B(\eta_0 A + B) \tag{44}$$

Hence the difference in the values in two choices is

$$Q_{\text{Red}} - Q_{\text{Blue}} = (\eta_0 A + B) \left( \frac{1}{2} \frac{1}{1 - \frac{1}{2}cA} - p_B \right). \tag{45}$$

Even in the case of $p_B > 1/2$, this expression can become positive by changing the time delay $T$ (*Figure 3B,C*), which can account for the irrational observing behaviors (*Spetch et al., 1990*; *Gipson et al., 2009*). The diagram in *Figure 3E* shows the results with the ansatz of *Equation (2)*, while *Figure 3—figure supplement 1* shows the results with the ansatz of *Equation (13)*. Note that the two ansatz provide qualitatively very similar results.

Since there are numerous variations of this experiment, here we provide with a formula for a more general case. Suppose a subject chooses either of two colored targets 100% info (I) or 0% info (N). If target I is chosen, one of the cues $S^+$ or $S^0$ is randomly presented with a probability of $p_I$ and $1 - p_I$, where $S^+$ is always followed by a reward with a size $R_I$ after time $T_{\text{Delay}}$, while $S^0$ is never followed by reward. If target N is chosen, a cue $S^*$ is presented and a reward with a size $R_2$ is followed after the fixed time delay $T_{\text{Delay}}$ with a probability of $p_{NI}$.

The difference in the values in two choices with the linear boosting ansatz is expressed as

$$Q_{\text{Info}} - Q_{\text{No-Info}} = \left( \frac{\eta_0}{\nu - \gamma}(e^{-\gamma T} - e^{-\nu T}) + e^{-\gamma T} \right) \left( \frac{p_I R_I}{1 - (1 - p_I)cA_I} - p_N R_N \right) \tag{46}$$

where,

$$A_I = \frac{R_I}{\nu - \gamma}(e^{-\gamma T} - e^{-\nu T}). \tag{47}$$

It is straightforward to apply this formula to specific experiments with a specific set of condition. For example, (*Stagner and Zentall, 2010*)'s experimental results can be accounted for by setting $R_I = R_N$, $p_I = 0.2$, and $p_N = 0.5$. As shown in *Figure 3—figure supplement 2*, our model reproduced the reported sub-optimal observing behavior.

## Model application 3: Our task on human participants

Our experiment, shown schematically in *Figure 4A*, was designed to test key aspects of the model. Subjects choose between info and non-info targets of values $Q_{\text{info}}$ and $Q_{\text{no-info}}$ respectively. The info target is followed by the reward predicting cue $S^+$ with the value of $Q^+$, followed by a reward $R^+$ after a delay of $T$, or the no-reward prediction cue $S^-$ (here we write $S^-$ instead of $S^0$ for our convention) with the value $Q^-$, followed by a no-reward with a value of $R^-$. The no-info target is followed by no cue (which we write $S^*$ for our convention) and randomly followed by a reward $R^+$ or no-reward $R^-$ after the delay of $T$. Note that we needed to introduce the value for the no-reward outcome based on the behaviors and self-reports.

Here, we fit the simplest form of the model to the subjects' behavior. Since we fit the model trial by trial, we need to introduce *learning*. After each trial, the value of chosen target was updated as

$$Q_X \rightarrow Q_X + \alpha(V - Q_X) \tag{48}$$

where $X =$info or no-info, $\alpha$ is the learning rate and $V$ is the reward function:

$$V = \eta_{S^i} A_i + B_i \tag{49}$$

with

$$A_i = \frac{R^i}{\nu_i - \gamma}\left(e^{-\gamma T} - e^{-\nu_i T}\right) \tag{50}$$

$$B_i = R^i e^{-\gamma T} \tag{51}$$

where $i = +$ or $-$, in case of a reward or a no-reward, respectively. We assumed that the anticipation rate $\nu^+$ and $\nu^-$ can be different but the discounting rate is the same $\gamma = \gamma^+ = \gamma^-$. The coefficient $\eta_{S^i}$ was assumed to be

$$\eta_{S^i} = \eta_0 + c\theta\left(\left|\delta_{pe}^{S_i}\right|\right). \tag{52}$$

This assumption allows us to reduce the number of free parameters. To see it, we write the expected values of targets:

$$Q_{\text{info}} = \frac{(\eta_0 + c)(A_+ + A_-) + B_+ + B_-}{2} \tag{53}$$

$$Q_{\text{no-info}} = \frac{\eta_0(A_+ + A_-) + B_+ + B_-}{2}. \tag{54}$$

Hence the expected difference is

$$\Delta Q = Q_{\text{info}} - Q_{\text{no-info}} = \frac{c(A_+ + A_-)}{2} \tag{55}$$

and the probability of choosing the info target $P_{\text{info}}$ is

$$P_{\text{info}} = \frac{1}{1 + e^{-\frac{\Delta Q}{\sigma}}}. \tag{56}$$

As is common, $c$ and $\sigma$ appear together with $R^i$'s ($cR^+/\sigma$, $cR^-/\sigma$). Thus, we set $c = 1$ and $\sigma = 1$ for the fitting. Thus the model has six free parameters that are fit: $R^+$, $R^-$, $\nu_+$, $\nu_-$, $\gamma$ and $\alpha$.

Note that a model that can allow asymmetric dependence of boosting on prediction errors (if it is positive or negative) leads to a related expression:

$$\eta_{S^i} \;=\; \eta_0 + c^+ \theta\!\left(\delta_{pe}^{S_i}\right) + c^- \theta\!\left(-\delta_{pe}^{S_i}\right), \tag{57}$$

which leads to

$$Q_{\text{info}} = \frac{\eta_0(A_+ + A_-) + c^+ A_+ + c^- A_- + B_+ + B_-}{2} \tag{58}$$

$$Q_{\text{no-info}} = \frac{\eta_0(A_+ + A_-) + B_+ + B_-}{2}. \tag{59}$$

Hence the expected difference is

$$\Delta Q = Q_{\text{info}} - Q_{\text{no-info}} = \frac{c^+ A_+ + c^- A_-}{2}. \tag{60}$$

Thus $c^+ R^+/\sigma$ and $c^- R^-/\sigma$ will be fitted as independent variables, which means that the asymmetric boosting will appear in the ratio between $R^+$, $R^-$.

For the purpose of model comparison, we also fitted the simple Q-learning model ($\gamma = \nu_+ = \nu_- = 0$), the Q-learning model with discounting ($\nu_+ = \nu_- = 0$), the RPE-boosting RL model with no value for the no-outcome ($R^- = 0$ and $\nu_- = 0$ ), the RPE-boosting RL model with no discounting ($\gamma = 0$), and the anticipation RL model with no-boosting. To fit the model with no-boosting ($c = 0$), we fitted a full model with $\eta_0$.

Note that from *Equation (60)*, the expected difference between the values of targets is zero $\Delta Q = 0$ when there is no boosting $c = 0$. This means that RPE-boosting is necessary to account for observing behaviors that prefer advance reward information.

## Acknowledgements

We would like to thank Ethan Bromberg-Martin, George Loewenstein, Kevin Lloyd, Mehdi Keramati for fruitful discussions, Molly Crockett for reward images with scoring data, and Elliot Ludvig for sharing his parallel studies. This work was supported by the Gatsby Charitable Foundation, the Wellcome Trust (091593/Z/10/Z, and Senior Investigator Award to RJD, 098362/Z/12/Z), the Joint Initiative on Computational Psychiatry and Ageing Research between the Max Planck Society and University College London (RJD)

## Additional information

### Funding

| Funder | Grant reference number | Author |
|---|---|---|
| Gatsby Charitable Foundation | | Kiyohito Iigaya<br>Peter Dayan |
| Wellcome Trust | 091593/Z/10/Z | Giles W Story<br>Zeb Kurth-Nelson<br>Raymond J Dolan |
| Wellcome Trust | 098362/Z/12/Z | Giles W Story<br>Zeb Kurth-Nelson<br>Raymond J Dolan |
| The Joint Initiative on Computational Psychiatry and Ageing Research between the Max Planck Society and University College London | | Giles W Story<br>Zeb Kurth-Nelson<br>Raymond J Dolan |

The funders had no role in study design, data collection and interpretation, or the decision to submit the work for publication.

## Author contributions
KI, Conception and design, Acquisition of data, Analysis and interpretation of data, Drafting or revising the article; GWS, ZK-N, PD, Conception and design, Analysis and interpretation of data, Drafting or revising the article; RJD, Analysis and interpretation of data, Drafting or revising the article

## Author ORCIDs
Kiyohito Iigaya, http://orcid.org/0000-0002-4748-8432
Peter Dayan, http://orcid.org/0000-0003-3476-1839

## Ethics
Human subjects: All participants provided written informed consent and consent to publish prior to start of the experiment, which was approved by the Research Ethics Committee at University College London (UCL Research Ethics Reference: 3450/002)

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
