## [Decision Letter]

Thank you for submitting your work entitled "The Modulation of Savouring by Prediction Error and its Effects on Choice" for consideration by *eLife*. Your article has been reviewed by three peer reviewers, and the evaluation has been overseen by Naoshige Uchida as the Reviewing Editor and Jody Culham as the Senior Editor. One of the three reviewers has agreed to reveal his identity: Sam Gershman.

The reviewers have discussed the reviews with one another and the Reviewing Editor has drafted this decision to help you prepare a revised submission.

Summary:

The authors present a model that is aimed to explain the following three behaviors:

1) "Information-seeking" behaviors and the observed neuronal activity in the lateral habenula in Bromberg-Martin and Hikosaka (2011).

2) The above "information-seeking" behavior becomes more prominent with longer delays to the extent that the animal chooses even the option that was associated with lower expected value (Spetch et al., 1990; Gipson et al., 2009).

3) Human data that the authors obtained (Figure 4).

The model has two key components:

1) Anticipation: The integral of the instantaneous anticipation function which grows over time (Figure 1, ii).

2) Boosting of anticipation by reward prediction error (Figure 1, iv).

Expected value of a cue at the time of cue presentation is determined by the sum of the (conventional) discounted reward value and the anticipation which is also temporally discounted (i.e. the instantaneous anticipation at future moments is more strongly temporally-discounted, Figure 1, iii). The model also makes other assumptions such as a linear relationship between the objective and subjective values of reward (which could affect risk preference of the model).

The authors conclude that their model can explain all the three data sets described above although the previous models were not able to explain these data. Although the question that the authors address is important, and the model is of potential interest, the reviewers have raised a number of concerns. Although the concerns regarding the authors' experiment are important points, the reviewers also noted that the main contribution of the present work could be the theory, and the modeling work can be of significance by itself if the authors perform more simulations and address the concerns described below. Therefore, the authors may choose to reduce the tone of the experimental part if it is difficult to address the concerns regarding the experiment.

Essential revisions:

1) It is unclear whether the data support the key prediction of a non-monotonic function of delay. In the legend of Figure 4, the authors indicate that "human subjects (n=14) showed a significant preference for the informative target for the case of long delays [20 sec: t(13) = 3.14, p=0.0078, 20 sec and 40 sec: t(27) = 4.00, p=0.00044]", and the three-star symbol (***) was placed between 20sec and 40sec delays. It is unclear whether these meant that the data supported one of the important predictions of the model (i.e. reversed preference with very long delays) or the comparison was between the short delays (2.5/7.5) versus pooled data for the two long delays (20/40). Please be more explicit. If the latter, the key prediction of a non-monotonic function of delay is not borne out by the data. This indicates that the model prediction is not supported by the data, at least for the range of delays tested.

2) The control experiment that had a different order of delays resulted in different results. This indicates that the result is sensitive to the ordering of trial types and might not be robust to experimental conditions.

3) Some of the comparisons involve the differing number of trials across conditions. The authors should correct this either by performing more experiments or by analyzing the data with a fixed number of trials.

4) The reviewers raised a number of concerns that the novelty over (or relationship with) previous studies is unclear. They pointed out specific literatures that the authors should refer to either basing on further simulations or with explicit discussions.

5) The model has several important assumptions, in particular, the two features described above (anticipation and boosting of anticipation by reward prediction errors). However, it is unclear what model features are essential in explaining the specific aspects of the data. What model features are important in explaining each of the three experiments? The authors should perform more explicit analyses addressing this question, for instance, by testing models with different components beyond that reported in Figure 4. Also, the robustness of the chosen model parameters should be reported further.

6) Reviewer #3 raised the issue that the predicted behavior of the model is suboptimal. Other referees appreciate, however, that even without a clear normative basis, the proposed model can be informative as far as the model explains a range of behaviors. However, it is sometimes the case that the behavior is suboptimal in specific conditions, yet the model performs more advantageously in different (perhaps more natural) conditions. If so, such performance could provide a reason for a given model. Please discuss this issue further.

7) Reviewer #3 was concerned that the model assumes perfect timing, and that the model might not be able to explain the behavior if it had timing that was realistic with respect to temporal uncertainty.

[Editors' note: further revisions were requested prior to acceptance, as described below.]

Thank you for resubmitting your work entitled "The Modulation of Savouring by Prediction Error and its Effects on Choice" for further consideration at *eLife*. Your revised article has been favorably evaluated by Jody Culham (Senior editor), a Reviewing editor, and three reviewers. The manuscript has been improved but there are some remaining issues that need to be addressed before acceptance, as outlined below:

Two reviewers (1 and 2) supported publication of your work but the other reviewer is still concerned that the present manuscript emphasizes the non-monotonicity of choice preference over delay too strongly. For instance, in the fourth paragraph of the subsection “Testing the preference for advance information about upcoming rewards across delays in human subjects” it is emphasized that the model predicted the inverted U-shape despite the experimental result not showing a significant drop with a longer delay. Overall, it would be more appropriate to reduce the tone, and discuss this issue as a limitation of the present model. One reviewer (3) remains unsatisfied about two issues. During discussion, the other reviewers noted, however, that incorporating all the details sometimes loses the power of modeling, and in this case, these issues can be dealt with in future investigations.

---

## [Author Response]

Essential revisions:

1) It is unclear whether the data support the key prediction of a non-monotonic function of delay. In the legend of Figure 4, the authors indicate that "human subjects (n=14) showed a significant preference for the informative target for the case of long delays [20 sec: t(13) = 3.14, p=0.0078, 20 sec and 40 sec: t(27) = 4.00, p=0.00044]", and the three-star symbol (***) was placed between 20sec and 40sec delays. It is unclear whether these meant that the data supported one of the important predictions of the model (i.e. reversed preference with very long delays) or the comparison was between the short delays (2.5/7.5) versus pooled data for the two long delays (20/40). Please be more explicit. If the latter, the key prediction of a non-monotonic function of delay is not borne out by the data. This indicates that the model prediction is not supported by the data, at least for the range of delays tested.

One of the predictions of our model is indeed a non-monotonic value dependence on delay. This has been confirmed by hypothetical questionnaire studies (Lowenstein, 1983). However, because we followed a design used in recent primate studies (Bromberg-Martin and Hikosaka, 2009, 2011), the reward probabilities of both targets were designed to be the same. Since a standard reinforcement-learning model predicts no preference in this design, we would not necessarily predict a preference reversal.

Nevertheless, the non-monotonic value functions that we hypothesize would be expected to have two consequences in our task. One is a finite interval of delays for which participants prefer approximately equally between info and no-info targets. This arises from the relative balance of savouring and dread, each of which has its own temporal characteristic, and both of which are boosted by prediction errors. Figure 4 shows how the various non-monotonic contributions can cancel each other, in our case for short delay conditions (T<10), the difference between the Q values of two targets were more similar to each other. This is due to the cancellation between the boosted savoring (the dotted-red line in the positive domain) and the boosted dread (the dotted-red line in the negative domain). Hence the choice probability should remain around 0.5. This is exactly what we found in our previous and new experiments (Figure 4, Figure 5), with the significant preference for 100% info target emerging only at longer delays of 20 sec and 40 sec. The sudden increase of preference was indeed due to the non-monotonic (discounting) dread function. This is now discussed explicitly in the Results:

“More precisely, for delays wherein the impact of savoring and dread were similarly strong, choice preference remained at around the chance level. […] By contrast, at longer delay conditions (> 20 sec), dread was discounted and savoring became dominant. This resulted in a large increase of preference for the 100% info target.”

Also, our additional model comparison analysis confirmed that the model with non-monotonic value functions (our original model with temporal discounting) outperformed the model with monotonic value functions (our model without temporal discounting). This is now stated:

“We stress that this sudden increase in choice preference was caused by the non-monotonic Q-value functions of targets (Figure 4). Our model comparison analysis also supported this conclusion (Figure 4), where the model with non-monotonic value functions (our original model with temporal discounting) outperformed the model with monotonic value functions (our model without temporal discounting).”

The second consequence of non-monotonicity is that, as pointed out by the reviewers, temporal discounting of savoring at extremely long delays should also make subjects become indifferent between info and non-info targets. Unfortunately, analysis of our model suggests that the delays concerned would be more than a few minutes – which was too long to be practicable in our design. Thus, we were not able to confirm this indifference in our task, and leave this for future studies. We now discuss this explicitly in the Results:

“Note that temporal discounting of savoring at extremely long delays should also make subjects indifferent between 100% and 0% info targets. […] Nonetheless, we stress that our model with temporal discounting outperformed a model without temporal discounting in terms of the iBIC scores (Figure 4).”

We apologize for the confusing comparisons in Figure 4, and have removed them. We now stress the significant preference of 100% info target at delay = 20 sec and 40 sec.

*2) The control experiment that had a different order of delays resulted in different results. This indicates that the result is sensitive to the ordering of trial types and might not be robust to experimental conditions.* We ran a new experiment in which the orders of the delays were randomized, and show this factor had no effect. The difference may arise out of the fact that for the control experiment in our original study participants were over-exposed to the same delay condition (>100 trials with 2.5 sec) before an abrupt change in delay (from 2.5 sec to 40 sec), while in our new experiment delay was randomized across trials and explicitly instructed by cues on each trial. We have added a comment about this to the Discussion:

“One issue that merits further future study is adaptation to different delays. […] Furthermore, if the prediction error could influence subjective time (for instance via a known effect of dopamine on aspects of timing (Gibbon et al. 1997)), then this could have complex additional effects on anticipation.”

*3) Some of the comparisons involve the differing number of trials across conditions. The authors should correct this either by performing more experiments or by analyzing the data with a fixed number of trials.* First, the new task involved the same number of trials in each condition, and again replicated the preference of 100% info target. Note that the original task was designed to equalize the amount of time in each condition – again with the same result.

Second, as suggested, we randomly sub-sampled the trials in our original experiment to equalize the number used per conditioning in calculating the statistics. This again confirmed our original result.

We now highlight both these facts in the Results and Methods:

“Also, Experiment-2 was designed to have an equal number of trials per delay condition, while Experiment-1 was designed to equalize the amount of time that participants spent in each condition (see Task Procedures in Materials and methods). The fact that we obtained the same results in both experiments illustrates the robustness of our findings.”

“Also, in order to treat different delay conditions equally, we randomly sub-sampled the trials to equalize the number used per condition in order to calculate the statistics. Additionally we obtained the same results by normalizing the posterior for each delay condition when estimating the expectation.”

*4) The reviewers raised a number of concerns that the novelty over (or relationship with) previous studies is unclear. They pointed out specific literatures that the authors should refer to either basing on further simulations or with explicit discussions.* First, we have attempted to make clearer the novelty of our suggestion – boosted anticipation and dread had never previously been considered – and also the relationship between our work and others. This led to changes in the Introduction and Discussion.

As suggested, we also applied our model to the task introduced by Zentall and colleagues involving comparison between 20% and 50% chance of rewards. Again, we could account for the reported suboptimal behavior (Figure 3—figure supplement 2). We have also provided with a general formula in the Methods section that can be applied to a wide range of experiments.

5) The model has several important assumptions, in particular, the two features described above (anticipation and boosting of anticipation by reward prediction errors). However, it is unclear what model features are essential in explaining the specific aspects of the data. What model features are important in explaining each of the three experiments? The authors should perform more explicit analyses addressing this question, for instance, by testing models with different components beyond that reported in Figure 4. Also, the robustness of the chosen model parameters should be reported further. We apologize that this was not clear in our previous version and we agree it is an absolutely critical point.

The assumption that is essential is that of boosting. Anticipation itself does not account for an advance information preference, as the advance information is irrelevant to the original formulation of anticipation of rewards (Loewenstain, 1987).

We conducted additional analysis of our model without boosting and showed how it behaves. In our behavioral task, and in the Bromberg-Martin task, the model predicts no preference without boosting (Figure 2—figure supplement 2); and a model restricted in this way fits our experimental data extremely poorly (76 iBIC units less than the model with boosting; Figure 4). In an experimental design from other labs in which behavior is known to be *suboptimal*, again, assuming no boosting leads, incorrectly, to *optimal* choice (Figure 3, Figure 3—figure supplement 2).

*6) Reviewer #3 raised the issue that the predicted behavior of the model is suboptimal. Other referees appreciate, however, that even without a clear normative basis, the proposed model can be informative as far as the model explains a range of behaviors. However, it is sometimes the case that the behavior is suboptimal in specific conditions, yet the model performs more advantageously in different (perhaps more natural) conditions. If so, such performance could provide a reason for a given model. Please discuss this issue further.* The observed preference for information is so strikingly suboptimal in a wide range of experiments, that it is hard to generate a plausibly normative explanation. Prediction errors have various untoward effects (for instance on happiness; Rutledge et al. 2014, 2015), leading to other non-normative consequences such as the hedonic treadmill. We now discuss non-normativity in more detail in Discussion:

“RPE-boosted anticipation, and like many apparently Pavlovian behaviors that involve innate responses, this appears evidently suboptimal – indeed as seen in strikingly non-normative choices. […] However, whether the behaviors reflect mechanistic constraints on neural computation (Kakade and Dayan, 2002), or a suitable adaptation to typical evolutionary environments, remains a question for further research.”

*7) Reviewer #3 was concerned that the model assumes perfect timing, and that the model might not be able to explain the behavior if it had timing that was realistic with respect to temporal uncertainty.*In all experiments that we analyzed in our paper, the delay was balanced between two targets. Thus the effect of timing uncertainty should be the same for both targets. Hence, we expect no effect on choice. We note this in the Discussion:

“In our task there was no effect of timing uncertainty on choice, as both targets are associated with the same delay. However it would become important if a task involves a choice between targets with different delay conditions. Furthermore, if the prediction error could influence subjective time (for instance via the known effect of dopamine on aspects of timing (Gibbon et al., 1997)), then this could have complex additional effects on anticipation.”

The reason that we assumed no dread in animal experiments is (as stated in Results) that:

“In these calculations, we set the value of no outcome to zero, implying a lack of dread in the no outcome condition. […] Moreover changing the delay between choice and cues that signaled no-reward had no impact on preference (McDevitt et al., 2016) Note, however, that our results still held in case of adding a finite value to the no outcome and this is something that we indeed found in our experiment, as detailed in the sections that follow.”

By contrast, we found that human participants assigned a negative value to no-outcome (Tobler et al., 2007), and therefore experienced dread. In fact, dread played a very important role in our task. We discuss this extensively for example in Results:

“In fitting the model, we were surprised to find that subjects assigned negative values to no-reward outcomes and its predicting cue (the bottom dotted line in Figure 4, see also Table 2) for the estimated model parameters). […] Omitting the negative value of the no-reward cue led to a failure to fit this effect (Figure 4—figure supplement 2).”

Also in Discussion:

“In our human experiments, we found that participants assigned a negative value to a no-outcome. […] Note that in the monkey experiments (Bromberg-Martin and Hikosaka, 2009, 2011), the lower-value outcome still involved an actual reward, albeit of a smaller size.”

The amount of dread was also boosted by reward prediction errors, hence the weighting function of anticipation (Eq.1) had to depend on an absolute value of prediction error.

With reference to reviewer #3’s concern about ‘absolute value’: we were using this in its purely mathematical form; that is the Euclidian distance from 0. For example, “the absolute value of −5 is 5”.

It is indeed a very good point that savoring and dread may not operate in the same dimension – we now refer to the important recent paper Fiorillo (2013) that considers this issue:

“It would be also interesting to study the difference between savouring and dread, as evidence shows that reward and punishment are not encoded in the same dimension (Fiorillo, 2013)”.

[Editors' note: further revisions were requested prior to acceptance, as described below.]

*The manuscript has been improved but there are some remaining issues that need to be addressed before acceptance, as outlined below: Two reviewers (1 and 2) supported publication of your work but the other reviewer is still concerned that the present manuscript emphasizes the non-monotonicity of choice preference over delay too strongly. For instance, in the fourth paragraph of the subsection “Testing the preference for advance information about upcoming rewards across delays in human subjects” it is emphasized that the model predicted the inverted U-shape despite the experimental result not showing a significant drop with a longer delay. Overall, it would be more appropriate to reduce the tone, and discuss this issue as a limitation of the present model. One reviewer (3) remains unsatisfied about two issues. During discussion, the other reviewers noted, however, that incorporating all the details sometimes loses the power of modeling, and in this case, these issues can be dealt with in future investigations.* Following your instructions, we made the suggested modest revisions.

In summary, we agree with your two major points. First, we appreciate Reviewer #1’s observation that we emphasized the non-monotonicity of choice preference over delay too strongly. We have implemented the suggestion to tone down this part of our discussion and claims, and indeed now cite it as a limitation of our current study.

Second, we agree with Reviewers #1 and #2’s suggestion that the issues raised by Reviewer #3’s can be most effectively addressed in future investigations. In fact, we think that some of Reviewer #3’s concerns are driven by a slight misunderstanding of our paper. This is because the majority of the issues that reviewer #3 raised in the review had in fact been addressed in our previous revision. We have therefore edited our manuscript to avoid any misreading.